# Semantic Density: Uncertainty Quantification for Large Language Models through Confidence Measurement in Semantic Space

**Xin Qiu**
Cognizant AI Labs
San Francisco, USA
qiuxin.nju@gmail.com

**Risto Miikkulainen**
Cognizant AI Labs, San Francisco, USA
The University of Texas at Austin, Austin, USA
risto@cognizant.com

## Abstract

With the widespread application of Large Language Models (LLMs) to various domains, concerns regarding the trustworthiness of LLMs in safety-critical scenarios have been raised, due to their unpredictable tendency to hallucinate and generate misinformation. Existing LLMs do not have an inherent functionality to provide the users with an uncertainty/confidence metric for each response it generates, making it difficult to evaluate trustworthiness. Although several studies aim to develop uncertainty quantification methods for LLMs, they have fundamental limitations, such as being restricted to classification tasks, requiring additional training and data, considering only lexical instead of semantic information, and being prompt-wise but not response-wise. A new framework is proposed in this paper to address these issues. Semantic density extracts uncertainty/confidence information for each response from a probability distribution perspective in semantic space. It has no restriction on task types and is "off-the-shelf" for new models and tasks. Experiments on seven state-of-the-art LLMs, including the latest Llama 3 and Mixtral-8x22B models, on four free-form question-answering benchmarks demonstrate the superior performance and robustness of semantic density compared to prior approaches.

## 1 Introduction

Large language models (LLMs) have revolutionized many domains, such as conversational agents [34], code generation [41], and mathematical discovery [39]. Given their ability for general reasoning and adaptability to new tasks, LLMs are utilized increasingly in safety-critical applications, including healthcare [42] and finance [50]. However, existing LLMs have an unpredictable tendency to hallucinate [30], leading to misleading information and risky behaviors. Responses are generated without quantitative indicators for their uncertainty/confidence, making it difficult to evaluate how trustworthy they are. As a result, concerns have been raised about their safety [54], hindering a deeper utilization of LLMs in risk-sensitive domains [5].

Although significant resources have been invested in LLM development, leading to a rapid pace in new model releases, only little progress has been made in building an uncertainty quantification framework for LLMs. An ideal outcome of such a system would be a quantitative metric associated with each response that can be used as an uncertainty/confidence indicator. Users can then build on this metric to evaluate the trustworthiness of LLM responses, e.g., establish an automatic system that triggers a warning if the response confidence is below a pre-defined threshold.

Following this line of thought, several techniques have been proposed in the literature to extract the uncertainty/confidence score from LLMs. In addition to the baselines that directly ask the LLM itself to evaluate its own answers [44, 21], one further step was to integrate traditional uncertainty estimation/calibration methods into LLMs [6, 51, 53]. However, due to the nature of these traditional methods, they only work on classification problems, not free-form natural language generation (NLG)

tasks, which are more general and challenging. Another direction was to fine-tune the original model [25] or train an additional layer or classifier [3, 28] to output uncertainty/confidence indicators for the responses. The main drawback is that these approaches are not "off-the-shelf" for new tasks and models: additional task-specific training labels of the ground-truth confidence are needed, and the training needs to be done in a model-specific manner, limiting their applicability.

Most importantly, prior work still treats LLM outputs as traditional auto-regressive predictions [29], i.e., the generated responses are simply handled as sequences of tokens/words, considering only their lexical uncertainty/confidence. However, due to the unique nature of free-form NLG, tokens that are lexically different may be semantically similar. In most LLM applications, decisions depend on the semantics of responses, and the same semantics can be stated using different words or sentence structures, leading to different lexical tokens. Therefore, uncertainty/confidence in semantic space is a more essential indicator for trustworthiness of LLM responses than lexical uncertainty/confidence.

Semantic entropy [22] is the state-of-the-art (SOTA) technique in semantic uncertainty [27]. However, its current design has two intrinsic limitations. First, the returned uncertainty score is prompt-wise, i.e., the semantic entropy is calculated for each prompt, instead of each response. Considering that LLMs can generate diverse responses for the same prompt, using the same uncertainty score for different responses is problematic [26]. Second, semantic entropy only considers semantic equivalence, which is a binary one-cut measurement, i.e., it only returns whether two responses are considered semantically equivalent or not, without reporting how semantically different the responses are. It does not make use of the more fine-grained semantic differences among the responses, which encode information that can make uncertainty quantification more precise.

To fill these gaps, a framework is developed in this paper for a new uncertainty/confidence metric, semantic density (SD), that can quantify the confidence of LLM responses in semantic space. Semantic density analyzes the output probability distribution from a semantic perspective and extracts a confidence indicator analogous to probability density. The proposed semantic density metric has the following advantages: (1) The returned confidence metric is response-wise, making it possible to evaluate the trustworthiness of each specific response; (2) it takes the fine-grained semantic differences among responses into account, which makes uncertainty quantification more precise; (3) it does not need any further training or fine-tuning of the original LLM; it is an "off-the-shelf" tool that can be directly applied to any pre-trained LLMs without modifying them; and (4) it does not pose any restrictions on the problem type; in particular, it works for general free-form generation tasks.

The performance of the semantic density metric was compared with six existing uncertainty/confidence quantification methods designed for LLMs across four question-answering benchmark datasets. All the approaches were tested on seven SOTA LLMs, including the latest Llama 3 and Mixtral-8x22B models. Semantic density performed significantly better than the alternatives across the board, suggesting that it forms a promising foundation for evaluating the trustworthiness of LLM responses. The source codes for reproducing the experimental results reported in this paper are provided at: https://github.com/cognizant-ai-labs/semantic-density-paper.

## 2    Related Work

In the LLM literature, the terms "uncertainty" and "confidence" are used in a mixed manner. A number of studies [51, 25, 12, 4, 16] treat uncertainty and confidence as two facets of a single concept, i.e., lower confidence on one particular response corresponds to higher uncertainty (or lower certainty). Other studies try to further differentiate "uncertainty" from "confidence" [26] and only use "uncertainty" to describe the entire output distribution instead of a specific response [22, 1]. Both perspectives fall in the same research area of "uncertainty quantification/estimation" [17, 26, 9, 16], and the goals of most existing uncertainty/confidence metrics are indeed the same: to provide a quantitative indicator of the trustworthiness of LLM responses. For better coverage and clarity, this paper uses "uncertainty quantification" as a general term to describe work related to the assessment of uncertainty or confidence of LLMs, and "uncertainty/confidence" to refer to multiple metrics with mixed term definitions. The proposed semantic density is thus an indicator of response-wise "confidence".

Although the main focus of the LLM community is still on developing new models with better performance, a number of studies aim at measuring uncertainty/confidence in LLMs. This section

summarizes their basic ideas and potential limitations, which are then targeted in the development of semantic density.

The first direction is to ask the LLM to evaluate the uncertainty/confidence of its own responses. Tian et al. [44] performed an empirical study showing that the inherent conditional probabilities are poorly calibrated in existing LLMs with RLHF (reinforcement learning from human feedback), and that verbal confidence estimates provided by the LLMs are better calibrated. Kadavath et al. [21] developed an approach where the LLM was asked to evaluate the correctness of its own answer, in the form of the probability "P(True)" that its own answer is correct. An additional "value" head was also trained to predict "P(True)", but it turned out not to generalize well to out-of-distribution datasets. In general, the performance of model self-evaluation is not as good as other more advanced uncertainty quantification methods [22].

A second direction is to integrate traditional uncertainty quantification methods into LLMs. The effectiveness of temperature scaling [13] for calibrating the output token probabilities of LLMs was verified in Desai and Durrett [6] and Xiao et al. [51]. Ye et al. [53] utilize conformal prediction to quantify the uncertainty of LLMs. However, these approaches are limited to NLP classification tasks; in contrast, the proposed semantic density can be applied to the more general and challenging free-form generation tasks.

A third direction is to perform supervised learning, either by fine-tuning the original LLM or adding an additional layer/classifier, to create uncertainty/confidence indicators. For example Lin et al. [25] fine-tuned the GPT-3 model to verbally express its own confidence level, and Azaria and Mitchell [3] trained an additional classifier to return the truthful probability of each generated response, based on the hidden layer activations of the LLM. Liu et al. [28] proposed the LitCab framework, in which a single linear layer is trained over the LLM's last hidden layer representations to predict a bias term, and the model's logits are then altered accordingly to update the response confidence. Since these methods are model-specific and need additional task-specific training labels, they cannot be readily applied to new models and tasks. In comparison, as an unsupervised method, semantic density is "off-the-shelf" for any new models and tasks, without the need for additional data or modifications to the original LLMs.

A common limitation of all the above methods is that they only consider the lexical information, and do not take into account the semantic relationships between responses. Yet semantics is critical in analyzing LLM outputs. As a fourth direction, semantic entropy [22] is a SOTA technique that quantifies semantic uncertainty for LLMs. It works by grouping the generated samples based on their semantic equivalence, and then generating an entropy-based indicator as an uncertainty metric. Although its performance is promising compared to the other approaches, as discussed in Section 1, it has two intrinsic limitations: (1) The generated semantic entropy is prompt-wise instead of response-wise, and (2) only a one-cut equivalence relationship is considered during semantic analysis. The second issue was considered in a recent follow-up [33] that tries to improve semantic entropy. However, it did not resolve the first issue, and the proposed uncertainty metric is still prompt-wise, limiting its utility when different responses are sampled given the same prompt. The proposed semantic density improves over these two aspects by providing a response-specific confidence indicator and analyzing the semantic relationship in a fine-grained manner.

Besides the above four major directions, uncertainty quantification for LLMs has been explored from other angles as well. Lin et al. [26] tested several simple baselines, among which a straightforward measurement of semantic dispersion is robust in evaluating selective response generation. Similarly, Manakul et al. [30] proposed a framework with several variants that use sampling consistency for detecting hallucinations. These two studies assume a very restricted condition in which the original sampling likelihoods of each response are not used; thus, the uncertainty/confidence information extracted by these methods is limited. Duan et al. [7] proposes a mechanism to shift attention to more relevant components at both token and sentence levels for better uncertainty quantification. Their approach was applied to prompt-wise uncertainty metrics only. The study by Ling et al. [27] focused on a specific in-context learning setting, aiming to decompose the uncertainty of LLMs into that caused by demonstration quality and that caused by model configuration. These experiments were limited to classification problems. Similarly, Hou et al. [15] decomposed the uncertainty into data uncertainty and model uncertainty in a prompt-wise approach. Xiao and Wang [52] studied the connections between hallucination and predictive uncertainty, showing that higher uncertainty is positively correlated with a higher chance of hallucination. This study validates the importance

of a reliable uncertainty measurement in detecting hallucinations of LLMs. Finally, Huang et al. [17] perform an explorative study using simple baselines on uncertainty measurement for LLMs, highlighting the need for more advanced uncertainty quantification methods developed exclusively for LLMs. This is the goal for the current paper as well.

## 3 Methodology

This section first defines the LLM uncertainty quantification problem, then describes the design principles and technical details of each algorithmic component, and concludes with a summary of the entire framework. The semantic space defined in Section 3.2 forms a theoretical foundation for semantic density. It can be implemented either explicitly through an embedding model or implicitly through an inference model; the latter is the approach used in this paper (details in Section 3.5).

### 3.1 Problem Statement

Given a pre-trained LLM, an input prompt $\boldsymbol{x}$, and an output sequence $\boldsymbol{y} = [y_1, y_2, \cdots, y_L]$, where $L$ is the number of tokens in $\boldsymbol{y}$, the target is to produce a confidence metric that is positively correlated with the probability of $\boldsymbol{y}$ to be true. Note that this metric should be response-wise, i.e., it is calculated for a specific $\boldsymbol{y}$ given $\boldsymbol{x}$. The metric can be used as a quantitative indicator for whether a specific response $\boldsymbol{y}$ can be trusted.

### 3.2 Semantic Space

Theoretically, a semantic space can be any *metric space* such that a distance function is properly defined to measure the semantic similarity between any two output responses, given the input prompt. Note that such a space is prompt-specific, i.e., each prompt results in a specific semantic space in which the distance function measures the contextual semantic similarity between two responses, treating the prompt as a common context.

More concretely, an oracle semantic space $\mathbb{S}$ is assumed to be a Euclidean space where each point is a $D$-dimensional vector that represents a contextual embedding of response $\boldsymbol{y}$ given prompt $\boldsymbol{x}$:

$$\boldsymbol{v} = E(\boldsymbol{y}|\boldsymbol{x}), \tag{1}$$

where $\boldsymbol{v} \in \mathbb{R}^D$, and $E(\cdot|\cdot)$ is an encoder that generates text embeddings with the following properties:

1. All the generated embedding vectors are normalized to have a norm of $\frac{1}{2}$:

$$||\boldsymbol{v}|| = \frac{1}{2}, \text{ for } \boldsymbol{v} = E(\boldsymbol{y}|\boldsymbol{x}), \ \forall \boldsymbol{x}, \boldsymbol{y}. \tag{2}$$

Whereas most existing text embedding models normalize the output vectors to have a norm of 1 [32], they are rescaled to $\frac{1}{2}$ without changing their direction to make it simpler to integrate them into the kernel function (as explained in Section 3.4).

2. Given a prompt $\boldsymbol{x}$ and two resulting responses $\boldsymbol{y}_i$ and $\boldsymbol{y}_j$, with $\boldsymbol{v}_i = E(\boldsymbol{y}_i|\boldsymbol{x})$ and $\boldsymbol{v}_j = E(\boldsymbol{y}_j|\boldsymbol{x})$, the following constraints exist for three extreme cases:

$$||\boldsymbol{v}_i - \boldsymbol{v}_j|| = \begin{cases} 0, \text{ if } \boldsymbol{y}_i \text{ and } \boldsymbol{y}_j \text{ are semantically equivalent given context } \boldsymbol{x} \\ \frac{\sqrt{2}}{2}, \text{ if } \boldsymbol{y}_i \text{ and } \boldsymbol{y}_j \text{ are semantically irrelevant given context } \boldsymbol{x} \\ 1, \text{ if } \boldsymbol{y}_i \text{ and } \boldsymbol{y}_j \text{ are semantically contradictory given context } \boldsymbol{x}. \end{cases} \tag{3}$$

Given the norm requirement in Eq. 2, the above three cases also correspond to $\boldsymbol{v}_i = \boldsymbol{v}_j$, $\boldsymbol{v}_i \perp \boldsymbol{v}_j$ and $\boldsymbol{v}_i = -\boldsymbol{v}_j$, respectively. Note that $||\boldsymbol{v}_i - \boldsymbol{v}_j||$ is not restricted to the above three values. It can be any value within $[0, 1]$, depending on the semantic similarity between $\boldsymbol{y}_i$ and $\boldsymbol{y}_j$ given $\boldsymbol{x}$.

3. Given a prompt $\boldsymbol{x}$ and three resulting responses $\boldsymbol{y}_i$, $\boldsymbol{y}_j$ and $\boldsymbol{y}_k$, with $\boldsymbol{v}_i = E(\boldsymbol{y}_i|\boldsymbol{x})$, $\boldsymbol{v}_j = E(\boldsymbol{y}_j|\boldsymbol{x})$ and $\boldsymbol{v}_k = E(\boldsymbol{y}_k|\boldsymbol{x})$,

$$||\boldsymbol{v}_i - \boldsymbol{v}_j|| < ||\boldsymbol{v}_i - \boldsymbol{v}_k||, \text{ if } \boldsymbol{y}_i \text{ is semantically closer to } \boldsymbol{y}_j \text{ than to } \boldsymbol{y}_k, \text{ given context } \boldsymbol{x}. \tag{4}$$

## 3.3 Semantic Density Estimator

Given the semantic space $\mathbb{S}$ defined in Section 3.2, the underlying probability distribution from which the LLM samples in $\mathbb{S}$ provides critical information: If a response is semantically close to many highly probable samples, it should be more trustworthy compared to a response that is semantically distant from the major sampling possibilities. A classical technique for estimating probability density is kernel density estimation (KDE) [40, 36]. However, the standard KDE only works for continuous variables, whereas the LLM outputs are discrete, i.e., sequences of tokens selected from a finite vocabulary. One possible way to extend KDE to accommodate LLM outputs is to build a density estimator as

$$\hat{p}(\boldsymbol{y}_*|\boldsymbol{x}) = \sum_{i=1}^{M} f_i K(\boldsymbol{v}_* - \boldsymbol{v}_i) = \frac{1}{\sum_{i=1}^{M} n_i} \sum_{i=1}^{M} n_i K(\boldsymbol{v}_* - \boldsymbol{v}_i), \tag{5}$$

where $\boldsymbol{x}$ is the input prompt, $\boldsymbol{y}_*$ is the target response, i.e., the response that needs confidence estimation, and $\boldsymbol{v}_* = E(\boldsymbol{y}_*|\boldsymbol{x})$. In total, $\sum_{i=1}^{M} n_i$ reference responses are sampled to facilitate the density estimation, where $M$ is the number of unique samples. Each $\boldsymbol{y}_i$ represents a unique sample; $n_i$ is the number of occurrences of $\boldsymbol{y}_i$ during sampling, $f_i = \frac{n_i}{\sum_{i=1}^{M} n_i}$ is the relative frequency of $\boldsymbol{y}_i$ during sampling, and $K(\cdot)$ is a kernel function.

The design of Eq. 5 is similar to an early variant of KDE [37] that was used to handle integer data. However, it has the drawback that it incorporates no knowledge about the sampling probabilities for each $\boldsymbol{y}_i$. It thus requires a large number of samplings, including a sufficient number of duplicated results, to obtain the relative frequency as an empirical approximation of the sampling probability. This cost becomes prohibitive for LLMs given how expensive LLM inference generally is.

In contrast with the inherently unknown probability distributions in standard KDE, the output token probabilities can be explicitly calculated in LLM sampling, and with this information, a more sample-efficient estimator can be developed. Given a prompt $\boldsymbol{x}$ and a resulting response $\boldsymbol{y}_*$, the semantic density of $\boldsymbol{y}_*$ is defined as

$$\text{SD}(\boldsymbol{y}_*|\boldsymbol{x}) = \frac{1}{\sum_{i=1}^{M} p(\boldsymbol{y}_i|\boldsymbol{x})} \sum_{i=1}^{M} p(\boldsymbol{y}_i|\boldsymbol{x}) K(\boldsymbol{v}_* - \boldsymbol{v}_i), \tag{6}$$

where $\boldsymbol{v}_* = E(\boldsymbol{y}_*|\boldsymbol{x})$, and $\boldsymbol{v}_i = E(\boldsymbol{y}_i|\boldsymbol{x})$ for $i = 1, 2, \cdots, M$. The $M$ unique responses $\boldsymbol{y}_i$ are the reference responses based on which the semantic density of $\boldsymbol{y}_*$ is estimated. $K(\cdot)$ is a kernel function which will be specified in Section 3.4, and $p(\boldsymbol{y}_i|\boldsymbol{x})$ (for $i = 1, 2, \cdots, M$) is the probability for the original LLM to generate sequence $\boldsymbol{y}_i$ given $\boldsymbol{x}$. That is, $p(\boldsymbol{y}_i|\boldsymbol{x}) = \prod_{j=1}^{L_i} p(y_{i,j}|y_{i,1}, y_{i,2}, \cdots, y_{i,j-1}, \boldsymbol{x})$, where $L_i$ is the number of tokens in $\boldsymbol{y}_i$ and $p(y_{i,j}|\cdot)$ is the conditional probability to generate token $y_{i,j}$. Note that in cases where $p(\boldsymbol{y}_*|\boldsymbol{x})$ is available, $\boldsymbol{y}_*$ can also be used as one of the $M$ reference responses.

One advantage of the semantic density estimator of Eq. 6 is that each result $\boldsymbol{y}_i$ only needs to be sampled once; their relative frequency $f_i$ can then be estimated as $f_i = \frac{p(\boldsymbol{y}_i|\boldsymbol{x})}{\sum_{i=1}^{M} p(\boldsymbol{y}_i|\boldsymbol{x})}$. Given a sampling budget of $M$ reference responses, it is therefore desirable that these $M$ samples are unique (duplications will be removed before calculating Eq. 6) and have high sampling probabilities, so that they can cover more sampling regions in the semantic space. In the current implementation, diverse beam search [46], which tends to generate diverse and highly probable responses, is used to sample the $M$ unique reference responses.

In practice, length-normalized probability [31, 29] is usually used to correct the length bias in sequence probability. Moreover, temperature scaling [13, 6, 51] is a simple yet effective method for calibrating the token probabilities during sampling. Both methods can be seamlessly integrated into semantic density: The $p(\boldsymbol{y}_i|\boldsymbol{x})$ in Eq. 6 can be replaced with $\sqrt[L_i]{p(\boldsymbol{y}_i|\boldsymbol{x})}$, and the temperature changed during sampling to calibrate each $p(y_{i,j}|\cdot)$.

## 3.4 Dimension-invariant Kernel

In a standard KDE setup, a commonly used kernel for multi-variate cases is the Epanechnikov kernel [8, 11], which was proved to be the most efficient in terms of asymptotic mean integrated squared

error [47]. Its original form is

$$K(\boldsymbol{v}) = \frac{\Gamma(2 + \frac{D}{2})}{\pi^{\frac{D}{2}}}(1 - ||\boldsymbol{v}||^2)\mathbf{1}_{||\boldsymbol{v}|| \leq 1}, \tag{7}$$

where $D$ is the dimension of vector $\boldsymbol{v}$, $\Gamma(\cdot)$ is the gamma function, $||\boldsymbol{v}||$ is the 2-norm of $\boldsymbol{v}$, and $\mathbf{1}_{condition}$ equals 1 if the condition is true, 0 otherwise.

In the semantic density estimator use case, one drawback of the original Epanechnikov kernel is that the normalization coefficient $\frac{\Gamma(2+\frac{D}{2})}{\pi^{\frac{D}{2}}}$ changes with the dimension $D$ of $\boldsymbol{v}$. As a result, semantic densities calculated using embeddings with different dimensionalities are incomparable. This issue may limit the flexibility in selecting embedding methodologies for semantic density calculation. However, the normalization coefficient can be removed to make the kernel function simpler and more flexible, without affecting the performance of confidence measurement. The kernel function of Eq. 6 thus becomes

$$K(\boldsymbol{v}_* - \boldsymbol{v}_i) = (1 - ||\boldsymbol{v}_* - \boldsymbol{v}_i||^2)\mathbf{1}_{||\boldsymbol{v}_* - \boldsymbol{v}_i|| \leq 1}. \tag{8}$$

Although the resulting kernel function does not meet the normalization requirement in standard KDE, it fits confidence estimation well. As long as the norm requirements in Eq. 2, 3 and 4 are fulfilled, any embedding models can be used to generate $\boldsymbol{v}$, regardless of the embedding dimensionalities. The outcome of the kernel function is always within $[0, 1]$, and a kernel value of 1, $\frac{1}{2}$, and 0 correspond to semantically equivalent, irrelevant, and contradictory responses, respectively. As a result, the semantic density is also within $[0, 1]$, with 1 as the highest semantic density a response can obtain, indicating that all the reference responses are semantically equivalent to it; analogously, it obtains a semantic density of 0 when all reference responses are semantically contradictory to it. This consistency in the value range makes practical applications of semantic density convenient: Practitioners can set a fixed threshold on semantic density to detect unreliable responses.

### 3.5 Semantic Distance Measurement via the Natural Language Inference (NLI) Model

Although most of the existing text-embedding models work in the semantic space defined in Section 3.2, they do not perform well in measuring semantic similarities [32]. Moreover, they can only consider input texts as a whole instead of doing a contextual encoding on part of the input, i.e., they can only obtain $E(\boldsymbol{x} + \boldsymbol{y})$ instead of $E(\boldsymbol{y}|\boldsymbol{x})$, where $\boldsymbol{x} + \boldsymbol{y}$ means a concatenation of $\boldsymbol{x}$ and $\boldsymbol{y}$.

The natural language inference (NLI) classification model[14] has proven to be effective in analyzing the semantic relationship between LLM responses with the prompt as context [22]. Given a pair of texts, an NLI model performs a classification task and outputs the probabilities for them to be semantically equivalent ("entailment" class), irrelevant ("neutral" class), or contradictory ("contradiction" class). Given the output class probabilities, the expectation of $||\boldsymbol{v}_* - \boldsymbol{v}_i||$ can be obtained as

$$\begin{aligned} \mathbb{E}(||\boldsymbol{v}_* - \boldsymbol{v}_i||^2) \quad &= 1^2 \cdot p_{\mathrm{c}}(\boldsymbol{y}_*, \boldsymbol{y}_i|\boldsymbol{x}) + (\tfrac{\sqrt{2}}{2})^2 \cdot p_{\mathrm{n}}(\boldsymbol{y}_*, \boldsymbol{y}_i|\boldsymbol{x}) + 0^2 \cdot p_{\mathrm{e}}(\boldsymbol{y}_*, \boldsymbol{y}_i|\boldsymbol{x}), \\ &= p_{\mathrm{c}}(\boldsymbol{y}_*, \boldsymbol{y}_i|\boldsymbol{x}) + \tfrac{1}{2} \cdot p_{\mathrm{n}}(\boldsymbol{y}_*, \boldsymbol{y}_i|\boldsymbol{x}) \end{aligned} \tag{9}$$

where $p_{\mathrm{c}}(\boldsymbol{y}_*, \boldsymbol{y}_i|\boldsymbol{x})$, $p_{\mathrm{n}}(\boldsymbol{y}_*, \boldsymbol{y}_i|\boldsymbol{x})$ and $p_{\mathrm{e}}(\boldsymbol{y}_*, \boldsymbol{y}_i|\boldsymbol{x})$ are the probabilities for $\boldsymbol{y}_*$ and $\boldsymbol{y}_i$ to be semantically contradictory ("c" for "contradiction" class), irrelevant ("n" for "neutral" class) and equivalent ("e" for "entailment" class), respectively, given context $\boldsymbol{x}$. During implementation, each response $\boldsymbol{y}$ will be concatenated with its prompt $\boldsymbol{x}$ (with prompt placed before the response) to form one text, i.e., $\boldsymbol{x} + \boldsymbol{y}$. Each input of the NLI model will then be a pair of these texts, analyzing the semantic relationship between two responses given the prompt. The expected value of $||\boldsymbol{v}_* - \boldsymbol{v}_i||^2$ can then be used in Eq. 8 to obtain the kernel function output.

### 3.6 Summary of the Semantic Density Framework

Algorithm 1 describes how the semantic density metric is deployed on a given task and model. The procedure consists of four main steps, i.e., sampling the reference responses, analyzing semantic relationships, calculating the kernel function, and calculating the semantic density.

**Computational Cost:** In terms of computational cost, only the first two steps involve model inferences. The first step utilizes diverse beam search, in which the group number equals $M$ with

---

**Algorithm 1** Procedure for deploying semantic density

---

**Require:**
    $\boldsymbol{y}_*$: target response that needs confidence measurement
    $\boldsymbol{x}$:  original prompt for generating $\boldsymbol{y}_*$
    $M$: number of unique reference responses to be sampled given $\boldsymbol{x}$
**Ensure:**
    $\mathrm{SD}(\boldsymbol{y}_*|\boldsymbol{x})$: semantic density for $\boldsymbol{y}_*$ given $\boldsymbol{x}$
**Step 1: Reference Response Sampling:**
1:  sample $M$ unique reference responses $\boldsymbol{y}_i$ (for $i = 1, 2, \cdots, M$) with prompt $\boldsymbol{x}$ on the original LLM using
    diverse beam search, and record each corresponding length-normalized sampling probability $\sqrt[L_i]{p(\boldsymbol{y}_i|\boldsymbol{x})}$
**Step 2: Semantic Relationship Analysis:**
2:  **for** $i = 1$ to $M$ **do**
3:     obtain $p_{\mathrm{c}}(\boldsymbol{y}_*, \boldsymbol{y}_i|\boldsymbol{x})$ and $p_{\mathrm{n}}(\boldsymbol{y}_*, \boldsymbol{y}_i|\boldsymbol{x})$ using NLI classification model
4:     calculate expectation $\mathbb{E}(||\boldsymbol{v}_* - \boldsymbol{v}_i||^2) = p_{\mathrm{c}}(\boldsymbol{y}_*, \boldsymbol{y}_i|\boldsymbol{x}) + \dfrac{1}{2} \cdot p_{\mathrm{n}}(\boldsymbol{y}_*, \boldsymbol{y}_i|\boldsymbol{x})$
**Step 3: Kernel Function Calculation:**
5:  **for** $i = 1$ to $M$ **do**
6:     calculate kernel function value using the expectation of $||\boldsymbol{v}_* - \boldsymbol{v}_i||^2$, given by:
    $K(\boldsymbol{v}_* - \boldsymbol{v}_i) = (1 - \mathbb{E}(||\boldsymbol{v}_* - \boldsymbol{v}_i||^2))\mathbf{1}_{\mathbb{E}(||\boldsymbol{v}_* - \boldsymbol{v}_i||) \leq 1}$
**Step 4: Semantic Density Calculation:**
7:  calculate semantic density: $\mathrm{SD}(\boldsymbol{y}_*|\boldsymbol{x}) = \dfrac{1}{\sum_{i=1}^{M} \sqrt[L_i]{p(\boldsymbol{y}_i|\boldsymbol{x})}} \sum_{i=1}^{M} \sqrt[L_i]{p(\boldsymbol{y}_i|\boldsymbol{x})} K(\boldsymbol{v}_* - \boldsymbol{v}_i)$

---

one beam in each group, and thus only $M$ inferences need to be done by the original LLM. The second step requires another $M$ or $2M$ inferences by the NLI classification model, depending on whether the relationship analysis is performed in a bi-directional manner. Considering the fact that NLI models are usually significantly smaller than LLMs (e.g., the Deberta-large-mnli model [14] used in the implementation in this paper only has 1.5 billion parameters), the computational cost is therefore mainly determined by the LLM inferences in the first step.

## 4   Experiments

This section first evaluates the performance of semantic density by comparing it with six existing uncertainty/confidence metrics over various LLMs and benchmarks. After that, two empirical studies are performed to investigate the robustness of semantic density when the number of reference responses and sampling strategy for target response are varied.

### 4.1   Performance Evaluation

Following the usual evaluation approach in the literature [22], the uncertainty/confidence metric is used as a quantitative indicator of how likely the response is going to be correct. Uncertainty values above a threshold are taken as incorrect while those below are taken as correct (and vice versa for confidence scores). For each threshold, the true positive rate vs. false positive rate is then measured. The area under this curve, namely area under receiver operator characteristic curve (AUROC), is calculated for each uncertainty/confidence metric. The AUROC score equals the probability that a randomly chosen incorrect response has a higher uncertainty than a randomly chosen correct response (and vice versa for confidence). A perfect uncertainty/confidence metric would have an AUROC score of 1 while a random metric would have 0.5. As an additional performance metric, the area under the Precision-Recall curve (AUPR) is also calculated. The average AUPR scores over two setups, i.e., using either correct or incorrect samples as the positive class, are reported. Note that a higher semantic density corresponds to a higher confidence.

The performance of semantic density (SD) was compared with six existing LLM uncertainty quantification methods: semantic entropy (SE) [22], P(True) [21], degree (Deg) [26], length-normalized likelihood (NL) [31], length-normalized entropy (NE) [29] and predictive entropy (PE) [21]. These methods were applied to seven SOTA open-source LLMs: Llama-2-13B [45], Llama-2-70B [45], Llama-3-8B [2], Llama-3-70B [2], Mistral-7B [18], Mixtral-8x7B [19] and Mixtral-8x22B [43]. Each LLM was tested on four free-form question-answering datasets commonly used in the literature: CoQA [38], TriviaQA [20], SciQ [48] and Natural Questions (NQ) [23]. For each question, 10

Table 1: Performance of different uncertainty/confidence metrics across various LLMs and datasets

| | CoQA | | | | | | |
| AUROC | SD | SE[22] | P(True)[21] | Deg[26] | NL[31] | NE[29] | PE[21] |
|---|---|---|---|---|---|---|---|
| Llama-2-13B | **0.783** | 0.633 | 0.594 | 0.734 | 0.709 | 0.629 | 0.647 |
| Llama-2-70B | **0.783** | 0.621 | 0.576 | 0.721 | 0.716 | 0.617 | 0.647 |
| Llama-3-8B | 0.738 | 0.599 | 0.593 | **0.795** | 0.676 | 0.608 | 0.604 |
| Llama-3-70B | **0.789** | 0.608 | 0.670 | 0.729 | 0.698 | 0.587 | 0.641 |
| Mistral-7B | **0.788** | 0.627 | 0.667 | 0.737 | 0.704 | 0.614 | 0.632 |
| Mixtral-8x7B | **0.786** | 0.626 | 0.589 | 0.728 | 0.708 | 0.617 | 0.651 |
| Mixtral-8x22B | **0.791** | 0.614 | 0.614 | 0.726 | 0.700 | 0.604 | 0.649 |

| | TriviaQA | | | | | | |
| AUROC | SD | SE | P(True) | Deg | NL | NE | PE |
|---|---|---|---|---|---|---|---|
| Llama-2-13B | **0.848** | 0.672 | 0.589 | 0.824 | 0.675 | 0.574 | 0.556 |
| Llama-2-70B | **0.829** | 0.677 | 0.556 | 0.787 | 0.714 | 0.582 | 0.566 |
| Llama-3-8B | **0.866** | 0.662 | 0.647 | 0.796 | 0.834 | 0.636 | 0.622 |
| Llama-3-70B | **0.828** | 0.663 | 0.654 | 0.764 | **0.828** | 0.611 | 0.596 |
| Mistral-7B | **0.866** | 0.690 | 0.589 | 0.828 | 0.745 | 0.615 | 0.536 |
| Mixtral-8x7B | **0.846** | 0.685 | 0.562 | 0.797 | 0.795 | 0.644 | 0.605 |
| Mixtral-8x22B | **0.829** | 0.686 | 0.604 | 0.762 | 0.801 | 0.644 | 0.607 |

| | SciQ | | | | | | |
| AUROC | SD | SE | P(True) | Deg | NL | NE | PE |
|---|---|---|---|---|---|---|---|
| Llama-2-13B | **0.757** | 0.570 | 0.572 | 0.727 | 0.693 | 0.513 | 0.574 |
| Llama-2-70B | **0.746** | 0.643 | 0.584 | 0.713 | 0.637 | 0.554 | 0.615 |
| Llama-3-8B | **0.780** | 0.611 | 0.564 | 0.731 | 0.686 | 0.597 | 0.651 |
| Llama-3-70B | **0.771** | 0.613 | 0.556 | 0.706 | 0.724 | 0.558 | 0.520 |
| Mistral-7B | **0.771** | 0.618 | 0.568 | 0.736 | 0.669 | 0.565 | 0.528 |
| Mixtral-8x7B | **0.773** | 0.612 | 0.585 | 0.716 | 0.726 | 0.612 | 0.658 |
| Mixtral-8x22B | **0.775** | 0.620 | 0.602 | 0.719 | 0.715 | 0.602 | 0.628 |

| | NQ | | | | | | |
| AUROC | SD | SE | P(True) | Deg | NL | NE | PE |
|---|---|---|---|---|---|---|---|
| Llama-2-13B | **0.689** | 0.581 | 0.592 | 0.686 | 0.588 | 0.571 | 0.640 |
| Llama-2-70B | 0.676 | 0.545 | 0.531 | **0.691** | 0.567 | 0.573 | 0.620 |
| Llama-3-8B | **0.710** | 0.583 | 0.517 | 0.706 | 0.601 | 0.603 | 0.615 |
| Llama-3-70B | **0.723** | 0.577 | 0.643 | 0.714 | 0.631 | 0.603 | 0.615 |
| Mistral-7B | **0.680** | 0.597 | 0.523 | 0.676 | 0.640 | 0.635 | 0.631 |
| Mixtral-8x7B | **0.729** | 0.599 | 0.576 | 0.720 | 0.654 | 0.603 | 0.608 |
| Mixtral-8x22B | **0.709** | 0.577 | 0.504 | 0.704 | 0.638 | 0.625 | 0.680 |

responses were generated using group beam search and used as reference responses in calculating SD, SE, Deg, NE, and PE (note that P(True) and NL do not need reference responses). Each unique response among these 10 will also be used as a target response, i.e., the response that needs an uncertainty/confidence estimation, in calculating the AUROC scores of uncertainty/confidence metrics. Detailed experimental configuration and parametric setup is described in Appendix A.1.

Table 1 and Table A2 (in Appendix A.4) show the AUROC and AUPR scores of each uncertainty/confidence metric across different models and datasets, with the best entry in each configuration highlighted in boldface. SD performs best in 26 out of 28 cases for AUROC and 27 out of 28 cases for AUPR, demonstrating that it is reliable and robust as a confidence metric for LLM responses. For AUROC, in two cases it is outperformed by Deg. After investigation, the inherent sequence likelihood returned by the original LLM was badly calibrated in these two cases. Deg is the only method that ignores the likelihood information during its calculation, making its performance unaffected by this negative factor. However, for the other 26 cases, SD is able to utilize the likelihood information to its advantage and outperform Deg. Appendix A.5 includes the results of another variant of SE [10], which is comparable to the original version. SD outperforms it in all the test cases.

To confirm that the observed performance differences in Table 1 are statistically significant, a paired $t$-test (paired by LLM and dataset) was performed between SD and the other metrics (Table A1 in

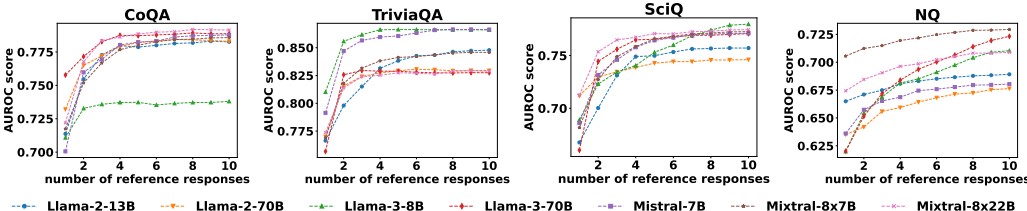

Figure 1: **AUROC scores of semantic density with one to 10 reference responses.** Each subfigure corresponds to the dataset indicated by the subfigure heading. Each curve corresponds to the base LLM identified in the legend below the subfigures. Providing more reference responses generally increases the reliability of semantic density, but in most cases only four samples are sufficient.

Appendix A.2). The $p$-values are consistently below $10^{-6}$, indicating that the performance gains of SD are strongly statistically significant.

The experiments reported above use the same setup as Kuhn et al. [22] to evaluate the correctness of responses: If the Rouge-L [24] between one response and the reference answer is larger than 0.3, then the response is deemed as correct. In order to investigate the influence of the correctness-checking criterion, the threshold of Rouge-L was varied between 0.1 and 1.0; the resulting performance of different uncertainty/confidence metrics are shown in Figure A1. SD consistently outperforms other methods under the different Rouge-L thresholds. Moreover, when the Rouge-L threshold increases, the AUROC scores generally increase for SD whereas they decrease for most other methods. Since a higher Rouge-L threshold means stricter correctness checking, the experimental performance gain of SD may be even larger if such checking is further improved (e.g. by using a more capable LLM).

To further evaluate the generalizability of SD in other types of tasks, an empirical study using a summarization task (DUC-2004 [35]) was performed. Appendix A.6 shows the performance comparisons among different uncertainty/confidence metrics. SD performs best in most test cases, demonstrating good generalizability.

### 4.2 Robustness of Semantic Density

Two additional empirical studies were performed to evaluate the robustness of semantic density when the number of reference responses varies or the sampling strategy for target response changes.

In the first study, the number of reference responses was reduced from 10, which is a standard setup for existing methods [22], to one, which is the extreme minimum case. Figure 1 shows the resulting AUROC scores, covering the same four datasets and seven LLMs. Although performance indeed decreases with fewer reference responses, the decrease is minor as long as the number of references is at least four. This result suggests that semantic density can provide reasonable performance even with a very limited budget for reference sampling.

In real-world applications, users may have different preferences when generating responses: Some may prefer a greedy sampling strategy while others may need diverse responses. The second study thus investigated how each uncertainty/confidence metric performs when the target response is sampled using different such strategies. The diverse beam search method inherently utilizes different strategies for each beam group: The first group performs a greedy beam search while later groups encourage more diverse responses. Following Section 4.1, the AUROC scores were calculated for target responses from each group separately, and the results averaged over the four datasets.

As the results in Figure 2 show, semantic density exhibits consistently good AUROC scores across different beam groups. Thus, it is robust against both more greedy and more diverse sampling strategies, thus covering a range of possible user preferences. In contrast, other approaches either perform consistently worse compared to semantic density across different beam groups, or their performance is unstable when the sampling strategy changes.

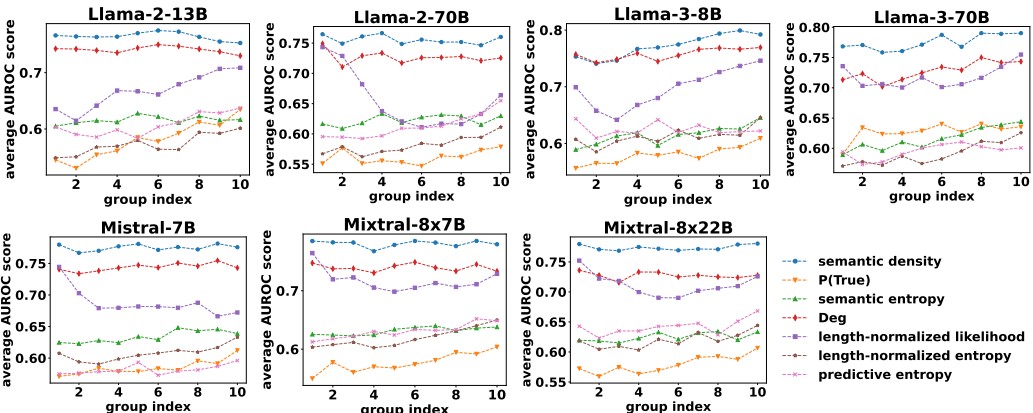

Figure 2: **AUROC scores over the different beam groups.** The plots show the average AUROC scores over the four datasets for the different beam groups in diverse beam search. A smaller group index corresponds to the group with a more greedy generation strategy, while the group with a larger index tends to be more diverse during response generation. Each subfigure corresponds to one of the seven LLMs, as indicated by the subfigure heading. Each curve represents one uncertainty/confidence metric indicated by the legend at the lower right. Semantic density exhibits consistently better and stable performance across different groups, compared to other methods.

## 5    Discussion and Future work

In terms of the broader societal impact of this work, the proposed semantic density provides a general way to evaluate the trustworthiness of responses generated by LLMs. This ability should have a positive impact on real-world applications that are safety-critical, such as healthcare and finance. Practitioners can utilize semantic density as an off-the-shelf indicator to filter out unreliable responses.

One limitation of semantic density is that it needs access to the output probabilities of generated tokens, which may not be available in some proprietary LLMs. In such a case, the more expensive variant in Eq. 5 can be considered as an alternative. Since semantic density does not require any further access to the internal states or weights of the original LLMs, it is still widely applicable. Another limitation is that most responses in the current experiments are at the sentence level. However, an extension of semantic density to long-paragraph responses should be feasible. Following the existing solutions [28, 10], a long response can be decomposed into sentence-level claims [28] or factoids [10], and semantic density then applied to estimate the confidence of each claim/factoid.

The framework for measuring semantic density is modular, and therefore its main components can be extended in future work. First, new sampling strategies that explicitly encourage a better coverage of semantic space can be developed to generate reference responses. Such extensions should improve the reliability of semantic density further. Second, text embedding methods that can measure contextual semantic similarity between responses more reliably will be helpful as well. Third, kernel functions specifically designed for the semantic space should allow the utilization of semantic relationships more efficiently. Fourth, more precise methods for calibrating inherent token probabilities will form a more reliable base for calculating semantic density.

## 6    Conclusion

This paper proposes semantic density as a practical new metric for measuring the confidence of LLM responses. It overcomes the limitations of existing approaches by utilizing the semantic information in an efficient and precise manner. It is response-specific, "off-the-shelf", and applicable to free-form generation tasks. Experimental comparisons with six existing uncertainty/confidence metrics across seven SOTA LLMs and four benchmark datasets suggest that it is accurate, robust, and general, and can therefore help deploy LLMs in safety-critical domains.

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

# A  Appendix

## A.1  Experimental Setup

This section provides the details of the experimental setup for reproducing the results presented in Section 4. The source codes for reproducing the reported experimental results are provided at: https://github.com/cognizant-ai-labs/semantic-density-paper.

**Base LLMs:** For all the seven base LLMs, the open-source versions in Huggingface Transformers library [49] were used in the experiments. More specifically, the following versions are used: `meta-llama/Llama-2-13b-hf`, `meta-llama/Llama-2-70b-hf`, `meta-llama/Meta-Llama-3-8B`, `meta-llama/Meta-Llama-3-70B`, `mistralai/Mistral-7B-v0.1`, `mistralai/Mixtral-8x7B-v0.1`, and `mistralai/Mixtral-8x22B-v0.1`. When running diverse beam search in generating the responses, the `generate()` function is used with `diversity_penalty=1.0`, a default `temperature` of 1.0, and the `num_beam_groups` equals the number of beams so that each group has exactly one beam.

**Correctness Checking:** Following the setup in Kuhn et al. [22], for all the reported experimental results except Figure A1, a response is considered to be correct if the Rouge-L [24] between it and any of the reference answers is larger than 0.3, after trimming the redundant continuations.

**Datasets:** The details of the datasets are as follows:

- **CoQA:** The `coqa-dev-v1.0` version is used with 1596 questions randomly selected for the experiments, using the Huggingface `datasets.train_test_split` function with a seed value of 10. The prompt format follows the same setup as in Kuhn et al. [22].

- **TriviaQA:** The dataset is loaded with the Huggingface `datasets` API. 1705 questions were randomly selected from the `validation` split for the experiments, using the Huggingface `datasets.train_test_split` function with a seed value of 10. A 10-shot prompt format is used following the setup in Kuhn et al. [22].

- **SciQ:** The `test` split from https://github.com/launchnlp/LitCab is used in the experiments. It contains 990 questions. A 10-shot prompt format is used following the setup in Kuhn et al. [22].

- **NQ:** The `test` split from https://github.com/launchnlp/LitCab is used in the experiments. 1800 questions were randomly selected using the Huggingface `datasets.train_test_split` function with a seed value of 10. A 10-shot prompt format is used following the setup in Kuhn et al. [22].

- **DUC-2004:** All the 500 samples in task 1 [35] is used in the experiments. The dataset is downloaded from https://www.kaggle.com/datasets/sandeep16064/duc2004. The following prompt is used for all the experiments: `"This is a bot that summarizes the following paragraph using one sentence. \n Paragraph: {document} \n Summary:"`.

**Uncertainty/Confidence Metrics:** The same exact target response and reference responses are used for all the tested methods. For SD, SE, and Deg, the `microsoft/deberta-large-mnli` model from Huggingface Transformers library [49] is used as the NLI classification model. Following Kuhn et al. [22], the probabilities for "contradiction", "neutral" and "entailment" are averaged bidirectionally. The parametric setup for the uncertainty metrics tested in Section 4 is summarized as follows:

- **semantic density (SD):** The same exact steps as described in Algorithm 1 were implemented, with a fixed temperature of 0.1 applied to rescale the value of each token probability during postprocessing.

- **semantic entropy (SE):** The original implementation and parametric setup from https://github.com/lorenzkuhn/semantic_uncertainty was used.

- **P(True):** The original few-shot prompt format from Kadavath et al. [21] is used.

- **degree (Deg):** The "entailment" probability returned by the NLI model (averaged bidirectionally) is used as the similarity between two responses, which is then used as the diagonal element in the degree matrix.

- **length-normalized likelihood (NL):** The original form as in Murray and Chiang [31] was implemented.

- **length-normalized entropy (NE):** The implementation from https://github.com/lorenzkuhn /semantic_uncertainty was used.

- **predictive entropy (PE):** The implementation from https://github.com/lorenzkuhn/semantic _uncertainty was used.

**Compute Resources:** All the experiments in Section 4 were running on an AWS P4de instance with 96 Intel(R) Xeon(R) Platinum 8275CL CPU @ 3.00GHz, 1152GB memory and 8 NVIDIA A100 (80GB). The GPU memory required to run the experiments is dependent on the base LLMs, as detailed below:

- **Llama-2-13B:** ∼30GB GPU memory.

- **Llama-2-70B:** ∼140GB GPU memory.

- **Llama-3-8B:** ∼20GB GPU memory.

- **Llama-3-70B:** ∼140GB GPU memory.

- **Mistral-7B:** ∼20GB GPU memory.

- **Mixtral-8x7B:** ∼110GB GPU memory.

- **Mixtral-8x22B:** ∼300GB GPU memory.

The exact computation time was affected by many factors, e.g., the current workload of the machine, the prompt length of the question, the generated response length, whether the cache option is turned on to store the model states of LLMs, etc.

## A.2 Results of Statistical Tests

Table A1 shows the $p$-values of the paired $t$-tests as described in Section 4.1.

Table A1: Statistical significance of SD's advantage over other methods ($p$-values of paired $t$-tests)

| SD vs. | SE | P(True) | Deg | NL | NE | PE |
|--------|------|---------|------|------|------|------|
| | 4.83E-17 | 1.71E-15 | 1.16E-7 | 4.62E-14 | 2.35E-15 | 4.63E-13 |

## A.3 Performances with Different Correctness Thresholds

Figure A1 shows the performance changes of the uncertainty/confidence metrics with different Rouge-L thresholds when checking the correctness of responses. SD performs the best overall across these different criteria.

## A.4 Performance Evaluation Using AUPR

During this evaluation, each uncertainty/confidence metric was used as a quantitative detector in two cases: 1) detecting incorrect responses, 2) detecting correct responses. The AUPR scores for both cases were calculated. Table A2 shows the average AUPR scores for them. Again, SD performs the best overall.

## A.5 Performance of Another Variant of Semantic Entropy

Table A3 shows the performance of a follow-up variant of semantic entropy [10]. The performance of this variant is comparable with the original semantic entropy, with several cases worse than the original version. The proposed semantic density outperforms this variant in all test cases.

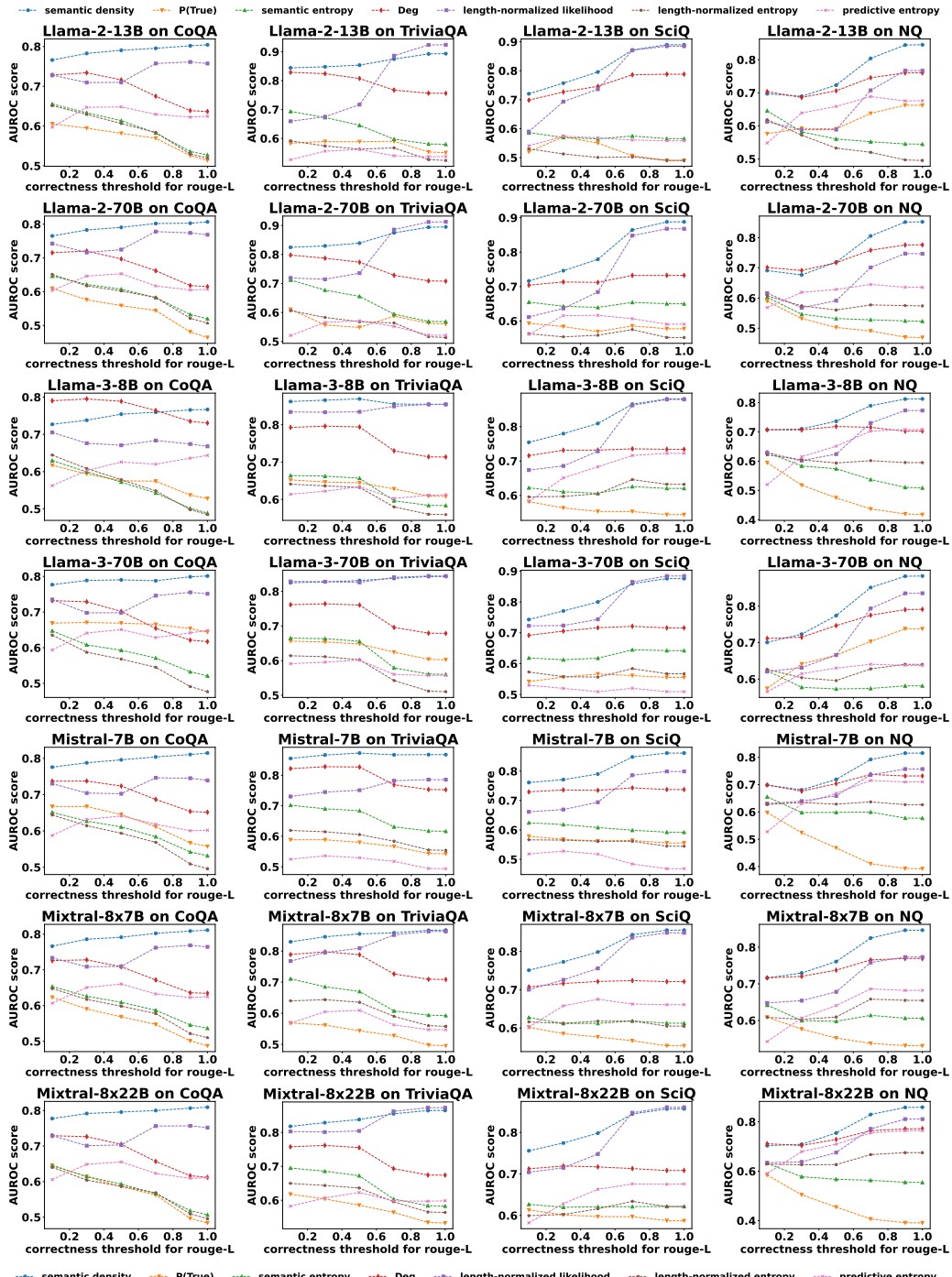

Figure A1: **Performance Evaluation over Different Correctness Checking Criteria.** The Rouge-L threshold for checking response correctness is set as 0.1, 0.3, 0.5, 0.7, 0.9 and 1.0, respectively. Each subfigure corresponds to one combination of base LLM and dataset, indicated by the subfigure heading. Each curve corresponds to the performance of one uncertainty/confidence metric identified in the legend at the very top and bottom. For each subfigure, the horizontal axis indicates the Rouge-L threshold, and the vertical axis indicates the AUROC score. Overall, semantic density exhibits best performance under different Rouge-L thresholds. The experimental performance of SD increases when the correctness-checking criterion becomes stricter (a higher Rouge-L threshold), while most other methods show opposite trends.

Table A2: Performance evaluation using AUPR-average metric

| CoQA | | | | | | | |
|---|---|---|---|---|---|---|---|
| AUPR | SD | SE | P(True) | Deg | NL | NE | PE |
| Llama-2-13B | **0.766** | 0.605 | 0.589 | 0.724 | 0.692 | 0.617 | 0.632 |
| Llama-2-70B | **0.766** | 0.587 | 0.566 | 0.710 | 0.686 | 0.602 | 0.628 |
| Llama-3-8B | 0.718 | 0.581 | 0.579 | **0.779** | 0.665 | 0.598 | 0.595 |
| Llama-3-70B | **0.765** | 0.572 | 0.641 | 0.714 | 0.658 | 0.566 | 0.606 |
| Mistral-7B | **0.773** | 0.600 | 0.652 | 0.730 | 0.680 | 0.599 | 0.618 |
| Mixtral-8x7B | **0.768** | 0.592 | 0.580 | 0.719 | 0.686 | 0.599 | 0.636 |
| Mixtral-8x22B | **0.772** | 0.581 | 0.605 | 0.715 | 0.672 | 0.585 | 0.632 |

| TriviaQA | | | | | | | |
|---|---|---|---|---|---|---|---|
| AUPR | SD | SE | P(True) | Deg | NL | NE | PE |
| Llama-2-13B | **0.839** | 0.651 | 0.589 | 0.807 | 0.662 | 0.570 | 0.550 |
| Llama-2-70B | **0.822** | 0.637 | 0.543 | 0.775 | 0.670 | 0.572 | 0.552 |
| Llama-3-8B | **0.855** | 0.636 | 0.636 | 0.781 | 0.815 | 0.638 | 0.616 |
| Llama-3-70B | **0.817** | 0.609 | 0.629 | 0.750 | 0.769 | 0.594 | 0.573 |
| Mistral-7B | **0.857** | 0.668 | 0.586 | 0.813 | 0.720 | 0.607 | 0.529 |
| Mixtral-8x7B | **0.835** | 0.653 | 0.546 | 0.782 | 0.763 | 0.628 | 0.591 |
| Mixtral-8x22B | **0.815** | 0.630 | 0.573 | 0.747 | 0.740 | 0.616 | 0.586 |

| SciQ | | | | | | | |
|---|---|---|---|---|---|---|---|
| AUPR | SD | SE | P(True) | Deg | NL | NE | PE |
| Llama-2-13B | **0.744** | 0.567 | 0.559 | 0.695 | 0.686 | 0.511 | 0.556 |
| Llama-2-70B | **0.733** | 0.626 | 0.574 | 0.694 | 0.630 | 0.549 | 0.604 |
| Llama-3-8B | **0.768** | 0.605 | 0.560 | 0.714 | 0.684 | 0.590 | 0.646 |
| Llama-3-70B | **0.755** | 0.597 | 0.551 | 0.690 | 0.722 | 0.548 | 0.518 |
| Mistral-7B | **0.759** | 0.612 | 0.559 | 0.717 | 0.664 | 0.561 | 0.530 |
| Mixtral-8x7B | **0.759** | 0.612 | 0.570 | 0.702 | 0.726 | 0.605 | 0.645 |
| Mixtral-8x22B | **0.753** | 0.604 | 0.588 | 0.703 | 0.700 | 0.589 | 0.628 |

| NQ | | | | | | | |
|---|---|---|---|---|---|---|---|
| AUPR | SD | SE | P(True) | Deg | NL | NE | PE |
| Llama-2-13B | **0.656** | 0.561 | 0.552 | 0.642 | 0.589 | 0.544 | 0.598 |
| Llama-2-70B | **0.655** | 0.539 | 0.525 | 0.648 | 0.580 | 0.547 | 0.591 |
| Llama-3-8B | **0.663** | 0.555 | 0.512 | 0.644 | 0.596 | 0.563 | 0.572 |
| Llama-3-70B | **0.700** | 0.562 | 0.608 | 0.655 | 0.638 | 0.570 | 0.580 |
| Mistral-7B | **0.654** | 0.573 | 0.519 | 0.637 | 0.623 | 0.586 | 0.586 |
| Mixtral-8x7B | **0.700** | 0.590 | 0.555 | 0.692 | 0.650 | 0.576 | 0.583 |
| Mixtral-8x22B | **0.687** | 0.571 | 0.509 | 0.671 | 0.645 | 0.596 | 0.647 |

Table A3: Performance of the semantic entropy variant

| AUROC | CoQA | TriviaQA | SciQ | NQ |
|---|---|---|---|---|
| Llama-2-13B | 0.587 | 0.726 | 0.576 | 0.576 |
| Llama-2-70B | 0.570 | 0.696 | 0.596 | 0.555 |
| Mistral-7B | 0.606 | 0.736 | 0.587 | 0.582 |
| Meta-Llama-3-8B | 0.571 | 0.694 | 0.582 | 0.563 |
| Meta-Llama-3-70B | 0.572 | 0.734 | 0.613 | 0.568 |
| Mixtral-8x7B | 0.567 | 0.699 | 0.576 | 0.612 |
| Mixtral-8x22B | 0.562 | 0.690 | 0.592 | 0.565 |

Table A4: Performance of different uncertainty metrics for summarization task

| AUROC | SD | DUC 2004 | | NL | NE | PE |
|---|---|---|---|---|---|---|
| | | SE | Deg | | | |
| Llama-2-13B | **0.622** | 0.612 | 0.606 | 0.594 | 0.601 | 0.521 |
| Llama-2-70B | 0.552 | **0.579** | 0.549 | 0.520 | 0.575 | 0.534 |
| Llama-3-8B | **0.597** | 0.529 | 0.590 | 0.538 | 0.538 | 0.521 |
| Llama-3-70B | **0.593** | 0.513 | 0.592 | 0.489 | 0.515 | 0.433 |
| Mistral-7B | **0.679** | 0.537 | 0.660 | 0.673 | 0.551 | 0.549 |
| Mixtral-8x7B | **0.640** | 0.532 | 0.620 | 0.636 | 0.576 | 0.497 |
| Mixtral-8x22B | **0.649** | 0.625 | 0.639 | 0.574 | 0.630 | 0.532 |

## A.6 Performace Evaluation in a Summarization Task

This experiment used Task 1 of DUC-2004 dataset [35], which is a single-document summarization task. The LLMs are prompted to summarize the document using one sentence. Table A4 presents the results. SD performs the best in six out of seven test cases, demonstrating its good generalizability to tasks other than question-answering.

