# OpenReview forum: "Semantic Density: Uncertainty Quantification for Large Language Models through Confidence Measurement in Semantic Space"
_NeurIPS.cc/2024/Conference — NeurIPS 2024 poster_

### Official Review · Reviewer_koQE · 2024-07-11

**Soundness:** 3
**Presentation:** 3
**Contribution:** 2
**Rating:** 5
**Confidence:** 4

**Summary:**

The paper introduces the metric "Semantic Density" to quantify the uncertainty of LLMs by measuring the distance of response embeddings in a semantic space.

**Strengths:**

- **Structure**: The problem is well-motivated and the paper is clearly and coherently written.
- **Theory**: The adaptation of kernel density estimation techniques to semantic space for uncertainty quantification in NLG is novel.
- **Experiments**: The experiments were performed on multiple datasets and various SOTA LLMs, where Semantic Density shows good empirical performance given the considered correctness metric.

**Weaknesses:**

- **Theory**: Semantic Density is theoretically not well grounded in uncertainty quantification theory, but proposed as an ad-hoc solution for practical application. So, although it seems intriguing to assign an uncertainty score response-wise rather than prompt-wise, uncertainty theory suggests that the (aleatoric semantic) uncertainty has to be quantified prompt-wise, no matter what ends up being sampled as the response [1]. However, since the true uncertainty score for a given prompt is unknown, the performance of an uncertainty measure is usually evaluated by how well the scores align with the correctness of the most-likely response.
- **Experiments**: According to the experimental setup in line 897, an answer is considered to be correct if its Rouge-L to any of the reference answers is larger than 0.3 for all datasets. However, since the considered datasets usually result in short responses, a fixed threshold of 0.3 is not sufficient. For instance, in the TriviaQA question *What is the last Grand Slam tennis tournament played in a calendar year?* the reference answer *The US Open* and the sampled response *The Australian Open* achieve a Rouge-L of approximately 0.66. This highlights the fact that considering multiple thresholds is essential to assess the performance of uncertainty metrics.

---
[1] L. Aichberger, K. Schweighofer, M. Ielanskyi, and S. Hochreiter. Semantically Diverse Language Generation for Uncertainty Estimation in Language Models. arXiv preprint arXiv:2406.04306, 2024.

**Questions:**

- What do the authors exactly mean by Semantic Density rebuilding the output probability distribution from a semantic perspective? (line 58)
- In line 277, the authors write "After investigation, the inherent sequence likelihood returned by the original LLM was badly calibrated in these two cases.". What is meant by badly calibrated sequence likelihoods? Why didn't the authors account for this by recomputing the sequence likelihood?
- There exists independent work (Kernel Language Entropy) that is closely related to Semantic Density [2]. The authors should consider discussing this in their paper.
- It would be insightful to additionally report the performance of existing uncertainty estimation methods when decreasing the number of reference responses instead of only reporting Semantic Density with different models (section 4.2)
- Semantic Density uses diverse beam search to sample responses, requiring hyperparameters such as diversity penalty and number of groups. How have these hyperparameters been chosen? Is the method sensitive to the specific sampling strategy? An ablation study on the performance of Semantic Density compared to existing uncertainty estimation methods would be insightful.
- Have the authors considered using sampling strategies other than diverse beam search? For instance, Aichberger et al. (2024) introduced sampling semantically diverse sequences, which could be utilized to sample unique reference responses [1].

---
[2] A. Nikitin, J. Kossen, Y. Gal, and P. Marttinen. Kernel Language Entropy: Fine-grained Uncertainty Quantification for LLMs from Semantic Similarities. arXiv preprint arXiv:2405.20003, 2024.

**Limitations:**

The authors did not address the limitations of their work, only the limitations of existing uncertainty estimation methods.

---

> ### Author Rebuttal · Authors · 2024-08-07
>
> Thanks for your constructive comments, we will answer your concerns follows the order of your original comments, due to the space limitation.
>
> ---
> Comment: concern regarding theory
>
> Response: Thanks for this insightful comment. We have read reference [1] carefully (it was published in arXiv only after the submission deadline, so we were not aware of it before), and will  address your concerns from two perspectives: First, considering the fact that LLMs sample responses stochastically given the same prompt, an uncertainty metric for each specific response has practical value. This is also reflected in reference [1], which  states “Uncertainty estimation in NLG involves assessing the uncertainty of an initially generated text (output sequence) for a given prompt (input sequence)”.The prompt-wise aleatoric uncertainty metric derived in [1] was indeed used to quantify the uncertainty of the most likely response (i.e.\ the initially generated response). While this estimate provides a good first approximation, assigning the same prompt-wise uncertainty score to different sampled responses is less useful in practical use cases where the trustworthiness of specific responses matters.The main motivation of the proposed semantic density is thus to provide a  solution for such use cases, i.e. to quantify the uncertainty of any specific response (not only the most likely one) that may be sampled by the LLM..
>
> Second, as an emerging research direction, uncertainty quantification in LLMs indeed needs more unified definitions of terms related to “uncertainty”. In the literature, different definitions and usages of the term “uncertainty” can be found, and sometimes other terms like “confidence” or “certainty” are used for the same purpose. Definitions of “uncertainty of a response” and “uncertainty of a prompt” should also be defined and differentiated. We will clarify that the “uncertainty” in this paper refers to “uncertainty of a response”, and delineate the different use cases. . We will also discuss reference [1] since we believe that it forms a good theoretical foundation for this emerging direction.
>
> ---
>
> Comment: concer regarding the experiments
>
> Response: Thanks for this constructive comment. As suggested, we have added experimental results using Rouge-L with different thresholds ranging from 0.1 to 1.0. According to the results, the performance gain of semantic density is not sensitive to the choice of Rouge-L thresholds.  The proposed semantic density consistently outperforms other methods across this range, further demonstrating that the results are robust.
>
> ---
>
> Response: Sorry for the confusion. We wanted to say that the original output distribution of LLMs is in sequence-of-token space, and we want to consider output distribution in a different semantic space. Theoretically, the output distribution in semantic space can be rebuilt from the output distribution in original sequence-of-token space. We will clarify this point in the revision.
>
> ---
>
> Response: Thanks for the question. Here, “badly calibrated sequence likelihoods” means the original likelihood of the LLM to sample the output token sequence is not well aligned with the correctness of the output sequence. We did not fine-tune or re-calibrate the original LLMs in our current experiments, because we wanted to evaluate the robustness of the proposed semantic density when working with existing LLMs in their original form. Post-hoc fine-tuning and calibration methods are complementary to semantic density, and they can always be applied to further improve the reliability of semantic density.
>
> ---
>
>
> Response: Thanks for the suggestion. We will include and discuss it in “related works”.
>
> ---
>
>
> Response: Thanks for the suggestion. We have added an experiment that investigates the performance change of semantic entropy when decreasing the number of reference responses. The patterns are similar for semantic entropy, but semantic density exhibits consistently better performance when the number of reference responses is reduced. We will add these results in the revision.
>
> ---
>
>
> Response: Thanks for the suggestion! We did not perform an extensive hyperparameter search for diverse beam search, and we used the standard default parameters. As suggested, we have added an ablation study following the setup in reference [1], which tests different choices of diversity penalty. Based on the results, a diversity penalty of 1.0 indeed provides the best performance, but the performance difference between 0.5 and 0.2 is not significant. We will add this empirical study to the revised version.
>
> ---
>
>
> Response: We have also tried standard multinomial sampling, and found that diverse beam search was better in generating unique and diverse generations. The SDLG sampling strategy in reference [1] is indeed a good match with semantic density, i.e., more diverse and likely reference responses would make semantic density more reliable. We will discuss it as a potential replacement for diverse beam search in the revision.
>
> ---
>
>
> Response: We have actually discussed the limitations of the method in several places in the original manuscript. Please see more details and justifications in section 2 of the appended NeurIPS Paper Checklist (page 15, after the references).

---

> > ### Comment · Reviewer_koQE · 2024-08-08
> >
> > Thank you for the rebuttal. However, I cannot locate any of the additional results you claim to have conducted for the rebuttal:
> >
> > > As suggested, we have added experimental results using Rouge-L with different thresholds ranging from 0.1 to 1.0.
> >
> > > We have added an experiment that investigates the performance change of semantic entropy when decreasing the number of reference responses.
> >
> > > As suggested, we have added an ablation study following the setup in reference [1], which tests different choices of diversity penalty.
> >
> > Could you please provide these results for further review?

---

> > > ### Author Response · Authors · 2024-08-09
> > > **Follow-up response to Reviewer koQE:**
> > >
> > > Thanks for spending time reading our rebuttals and providing further comments! We did not include the detailed results in the initial rebuttal because this year Neurips only allows 6,000 characters in the initial rebuttal to each reviewer, urging the authors to be more concise. Moreover, only one page of PDF can be attached in the general rebuttal to include necessary tables/figures, and no external URL is allowed. Given this restriction, we have to conclude the additional experimental results with a very brief text summary in the initial rebuttal. However, we understand that providing the detailed results would be more informative to you, so we will attach all the detailed results in follow-up comments per your request. Due to the format limitation, we will use markdown tables to display all the results. Please check them in the comments following this one. We also provide some further discussions below:
> > >
> > >
> > > The most interesting results are indeed the Rouge-L threshold study that you suggested as one of your main comments. In additional to the observation that the proposed semantic density still provides overall best performance across all the tested Rouge-L thresholds, we also noticed that the absolute AUROC scores show an increasing trend for semantic density when the Rouge-L threshold is increased. This indicates that the strictness of correctness check is indeed affecting the absolute performance of the evaluated methods. The initial Rouge-L threshold of 0.3 used in the experiments was following the same setup in the original semantic entropy paper [3], aiming to make the comparisons fair. However, we agree that it is indeed more informative to include results that are under different Rouge-L thresholds. We will include all the results in the revision, and add a discussion regarding this aspect. Thanks for this constructive comment!
> > >
> > >
> > > [3] Lorenz Kuhn and Yarin Gal and Sebastian Farquhar, “Semantic Uncertainty: Linguistic Invariances for Uncertainty Estimation in Natural Language Generation”, The Eleventh International Conference on Learning Representations (ICLR), 2023

---

> ### Author Response · Authors · 2024-08-09
> **Detailed results for Rouge-L threshold of 0.1 and 0.5**
>
> Below are the AUROC scores when using a Rouge-L threshold of 0.1 and 0.5 (0.3 has already been shown in the original manuscript):
>
> ## Results for Rouge-L threshold of 0.1:
>
> ### **CoQA**
> | Rouge-L >= 0.1 | SD | SE | P(True) | Deg | NL | NE | PE |
> |----------------|----|----|---------|-----|----|----|----|
> |Llama-2-13b-hf|0.766|0.623|0.606|0.728|0.702|0.556|0.508|
> |Llama-2-70b-hf|0.765|0.622|0.610|0.715|0.719|0.572|0.538|
> |Meta-Llama-3-8B|0.727|0.601|0.617|0.790|0.632|0.566|0.511|
> |Meta-Llama-3-70B|0.777|0.606|0.669|0.732|0.712|0.542|0.489|
> |Mistral-7B-v0.1|0.776|0.616|0.667|0.737|0.705|0.553|0.508|
> |Mixtral-8x7B-v0.1|0.766|0.624|0.623|0.726|0.707|0.555|0.520|
> |Mixtral-8x22B-v0.1|0.777|0.617|0.646|0.729|0.710|0.560|0.525|
> ### **TriviaQA**
> | Rouge-L >= 0.1 | SD | SE | P(True) | Deg | NL | NE | PE |
> |----------------|----|----|---------|-----|----|----|----|
> |Llama-2-13b-hf|0.844|0.632|0.583|0.829|0.640|0.505|0.490|
> |Llama-2-70b-hf|0.824|0.664|0.611|0.798|0.696|0.516|0.472|
> |Meta-Llama-3-8B|0.862|0.572|0.652|0.793|0.798|0.480|0.425|
> |Meta-Llama-3-70B|0.825|0.624|0.656|0.762|0.804|0.520|0.491|
> |Mistral-7B-v0.1|0.855|0.620|0.589|0.822|0.693|0.487|0.476|
> |Mixtral-8x7B-v0.1|0.830|0.654|0.569|0.789|0.742|0.541|0.511|
> |Mixtral-8x22B-v0.1|0.818|0.660|0.618|0.758|0.779|0.577|0.535|
> ### **Sciq**
> | Rouge-L >= 0.1 | SD | SE | P(True) | Deg | NL | NE | PE |
> |----------------|----|----|---------|-----|----|----|----|
> |Llama-2-13b-hf|0.720|0.570|0.520|0.698|0.575|0.458|0.500|
> |Llama-2-70b-hf|0.716|0.639|0.593|0.704|0.576|0.476|0.507|
> |Meta-Llama-3-8B|0.755|0.583|0.582|0.716|0.633|0.468|0.526|
> |Meta-Llama-3-70B|0.743|0.588|0.542|0.692|0.694|0.490|0.495|
> |Mistral-7B-v0.1|0.761|0.588|0.578|0.729|0.635|0.462|0.468|
> |Mixtral-8x7B-v0.1|0.751|0.598|0.602|0.707|0.675|0.523|0.566|
> |Mixtral-8x22B-v0.1|0.756|0.600|0.613|0.712|0.677|0.500|0.525|
> ### **NQ**
> | Rouge-L >= 0.1 | SD | SE | P(True) | Deg | NL | NE | PE |
> |----------------|----|----|---------|-----|----|----|----|
> |Llama-2-13b-hf|0.697|0.589|0.576|0.704|0.555|0.521|0.503|
> |Llama-2-70b-hf|0.691|0.553|0.590|0.701|0.555|0.506|0.488|
> |Meta-Llama-3-8B|0.707|0.536|0.595|0.707|0.553|0.466|0.444|
> |Meta-Llama-3-70B|0.701|0.573|0.574|0.711|0.556|0.503|0.490|
> |Mistral-7B-v0.1|0.698|0.574|0.598|0.699|0.558|0.493|0.454|
> |Mixtral-8x7B-v0.1|0.716|0.576|0.609|0.716|0.588|0.503|0.484|
> |Mixtral-8x22B-v0.1|0.703|0.579|0.584|0.711|0.575|0.518|0.517|
>
> ## Results for Rouge-L threshold of 0.5:
>
> ### **CoQA**
> | Rouge-L >= 0.5 | SD | SE | P(True) | Deg | NL | NE | PE |
> |----------------|----|----|---------|-----|----|----|----|
> |Llama-2-13b-hf|0.791|0.598|0.582|0.716|0.701|0.503|0.515|
> |Llama-2-70b-hf|0.791|0.599|0.559|0.697|0.712|0.506|0.527|
> |Meta-Llama-3-8B|0.754|0.558|0.575|0.789|0.633|0.500|0.523|
> |Meta-Llama-3-70B|0.790|0.585|0.669|0.702|0.694|0.470|0.494|
> |Mistral-7B-v0.1|0.796|0.599|0.645|0.724|0.695|0.495|0.514|
> |Mixtral-8x7B-v0.1|0.791|0.598|0.568|0.709|0.700|0.503|0.530|
> |Mixtral-8x22B-v0.1|0.796|0.593|0.586|0.705|0.699|0.505|0.538|
> ### **TriviaQA**
> | Rouge-L >= 0.5 | SD | SE | P(True) | Deg | NL | NE | PE |
> |----------------|----|----|---------|-----|----|----|----|
> |Llama-2-13b-hf|0.854|0.593|0.588|0.807|0.689|0.479|0.518|
> |Llama-2-70b-hf|0.839|0.620|0.548|0.773|0.710|0.491|0.519|
> |Meta-Llama-3-8B|0.869|0.562|0.645|0.794|0.799|0.464|0.428|
> |Meta-Llama-3-70B|0.832|0.616|0.648|0.761|0.801|0.504|0.489|
> |Mistral-7B-v0.1|0.873|0.606|0.580|0.827|0.712|0.476|0.472|
> |Mixtral-8x7B-v0.1|0.855|0.623|0.544|0.788|0.767|0.546|0.549|
> |Mixtral-8x22B-v0.1|0.838|0.641|0.585|0.755|0.780|0.558|0.569|
> ### **SciQ**
> | Rouge-L >= 0.5 | SD | SE | P(True) | Deg | NL | NE | PE |
> |----------------|----|----|---------|-----|----|----|----|
> |Llama-2-13b-hf|0.796|0.557|0.550|0.746|0.704|0.418|0.511|
> |Llama-2-70b-hf|0.779|0.644|0.569|0.712|0.649|0.460|0.548|
> |Meta-Llama-3-8B|0.809|0.575|0.554|0.732|0.693|0.449|0.613|
> |Meta-Llama-3-70B|0.800|0.601|0.567|0.716|0.716|0.452|0.454|
> |Mistral-7B-v0.1|0.790|0.578|0.563|0.735|0.669|0.433|0.443|
> |Mixtral-8x7B-v0.1|0.798|0.599|0.576|0.721|0.732|0.507|0.618|
> |Mixtral-8x22B-v0.1|0.798|0.603|0.597|0.717|0.719|0.487|0.584|
> ### **NQ**
> | Rouge-L >= 0.5 | SD | SE | P(True) | Deg | NL | NE | PE |
> |----------------|----|----|---------|-----|----|----|----|
> |Llama-2-13b-hf|0.723|0.494|0.591|0.706|0.528|0.392|0.496|
> |Llama-2-70b-hf|0.719|0.487|0.503|0.717|0.534|0.438|0.486|
> |Meta-Llama-3-8B|0.736|0.487|0.475|0.718|0.589|0.419|0.535|
> |Meta-Llama-3-70B|0.774|0.545|0.666|0.747|0.641|0.455|0.524|
> |Mistral-7B-v0.1|0.719|0.518|0.468|0.703|0.583|0.441|0.511|
> |Mixtral-8x7B-v0.1|0.761|0.526|0.552|0.737|0.611|0.452|0.510|
> |Mixtral-8x22B-v0.1|0.755|0.526|0.455|0.729|0.607|0.473|0.584|

---

> > ### Author Response · Authors · 2024-08-09
> > **Detailed results for Rouge-L threshold of 0.7 and 0.9**
> >
> > ## Results for Rouge-L threshold of 0.7:
> >
> > ### **CoQA**
> > | Rouge-L >= 0.7 | SD | SE | P(True) | Deg | NL | NE | PE |
> > |----------------|----|----|---------|-----|----|----|----|
> > |Llama-2-13b-hf|0.796|0.577|0.569|0.675|0.767|0.494|0.509|
> > |Llama-2-70b-hf|0.802|0.576|0.544|0.662|0.780|0.493|0.503|
> > |Meta-Llama-3-8B|0.759|0.537|0.574|0.764|0.667|0.481|0.526|
> > |Meta-Llama-3-70B|0.788|0.570|0.665|0.654|0.754|0.456|0.485|
> > |Mistral-7B-v0.1|0.804|0.581|0.612|0.687|0.759|0.481|0.501|
> > |Mixtral-8x7B-v0.1|0.802|0.583|0.547|0.672|0.768|0.486|0.508|
> > |Mixtral-8x22B-v0.1|0.800|0.565|0.562|0.657|0.763|0.492|0.515|
> > ### **TriviaQA**
> > | Rouge-L >= 0.7 | SD | SE | P(True) | Deg | NL | NE | PE |
> > |----------------|----|----|---------|-----|----|----|----|
> > |Llama-2-13b-hf|0.875|0.559|0.590|0.767|0.869|0.505|0.513|
> > |Llama-2-70b-hf|0.874|0.572|0.587|0.728|0.871|0.514|0.524|
> > |Meta-Llama-3-8B|0.856|0.517|0.628|0.730|0.827|0.432|0.417|
> > |Meta-Llama-3-70B|0.838|0.552|0.624|0.696|0.821|0.458|0.461|
> > |Mistral-7B-v0.1|0.867|0.568|0.567|0.769|0.761|0.474|0.470|
> > |Mixtral-8x7B-v0.1|0.858|0.575|0.528|0.726|0.826|0.516|0.517|
> > |Mixtral-8x22B-v0.1|0.855|0.584|0.564|0.693|0.846|0.531|0.556|
> > ### **SciQ**
> > | Rouge-L >= 0.7 | SD | SE | P(True) | Deg | NL | NE | PE |
> > |----------------|----|----|---------|-----|----|----|----|
> > |Llama-2-13b-hf|0.872|0.577|0.507|0.786|0.860|0.422|0.507|
> > |Llama-2-70b-hf|0.864|0.671|0.586|0.732|0.835|0.481|0.548|
> > |Meta-Llama-3-8B|0.865|0.602|0.553|0.735|0.855|0.484|0.652|
> > |Meta-Llama-3-70B|0.860|0.641|0.561|0.721|0.857|0.474|0.463|
> > |Mistral-7B-v0.1|0.848|0.578|0.564|0.743|0.785|0.432|0.415|
> > |Mixtral-8x7B-v0.1|0.843|0.610|0.567|0.724|0.832|0.508|0.608|
> > |Mixtral-8x22B-v0.1|0.844|0.614|0.597|0.713|0.843|0.508|0.600|
> > ### **NQ**
> > | Rouge-L >= 0.7 | SD | SE | P(True) | Deg | NL | NE | PE |
> > |----------------|----|----|---------|-----|----|----|----|
> > |Llama-2-13b-hf|0.804|0.494|0.638|0.746|0.662|0.378|0.518|
> > |Llama-2-70b-hf|0.806|0.489|0.491|0.758|0.662|0.432|0.488|
> > |Meta-Llama-3-8B|0.789|0.454|0.437|0.715|0.742|0.423|0.584|
> > |Meta-Llama-3-70B|0.851|0.557|0.703|0.775|0.784|0.471|0.532|
> > |Mistral-7B-v0.1|0.792|0.527|0.410|0.737|0.694|0.441|0.544|
> > |Mixtral-8x7B-v0.1|0.825|0.542|0.538|0.765|0.704|0.483|0.550|
> > |Mixtral-8x22B-v0.1|0.829|0.530|0.407|0.764|0.722|0.501|0.640|
> >
> > ## Results for Rouge-L threshold of 0.9:
> > ### **CoQA**
> > | Rouge-L >= 0.9 | SD | SE | P(True) | Deg | NL | NE | PE |
> > |----------------|----|----|---------|-----|----|----|----|
> > |Llama-2-13b-hf|0.802|0.553|0.525|0.639|0.797|0.441|0.477|
> > |Llama-2-70b-hf|0.803|0.553|0.481|0.618|0.800|0.436|0.470|
> > |Meta-Llama-3-8B|0.766|0.505|0.537|0.735|0.684|0.432|0.514|
> > |Meta-Llama-3-70B|0.799|0.568|0.654|0.621|0.786|0.401|0.471|
> > |Mistral-7B-v0.1|0.811|0.566|0.567|0.654|0.784|0.426|0.466|
> > |Mixtral-8x7B-v0.1|0.809|0.568|0.501|0.636|0.798|0.432|0.477|
> > |Mixtral-8x22B-v0.1|0.807|0.544|0.496|0.617|0.786|0.433|0.478|
> > ### **TriviaQA**
> > | Rouge-L >= 0.9 | SD | SE | P(True) | Deg | NL | NE | PE |
> > |----------------|----|----|---------|-----|----|----|----|
> > |Llama-2-13b-hf|0.892|0.547|0.553|0.756|0.913|0.468|0.508|
> > |Llama-2-70b-hf|0.894|0.559|0.563|0.709|0.902|0.468|0.491|
> > |Meta-Llama-3-8B|0.855|0.508|0.609|0.714|0.838|0.414|0.417|
> > |Meta-Llama-3-70B|0.843|0.541|0.604|0.679|0.828|0.426|0.447|
> > |Mistral-7B-v0.1|0.868|0.559|0.544|0.753|0.772|0.449|0.447|
> > |Mixtral-8x7B-v0.1|0.867|0.569|0.498|0.709|0.844|0.489|0.503|
> > |Mixtral-8x22B-v0.1|0.865|0.571|0.535|0.674|0.860|0.497|0.547|
> > ### **SciQ**
> > | Rouge-L >= 0.9 | SD | SE | P(True) | Deg | NL | NE | PE |
> > |----------------|----|----|---------|-----|----|----|----|
> > |Llama-2-13b-hf|0.890|0.574|0.490|0.788|0.880|0.407|0.503|
> > |Llama-2-70b-hf|0.888|0.676|0.577|0.732|0.861|0.453|0.526|
> > |Meta-Llama-3-8B|0.881|0.602|0.544|0.734|0.881|0.464|0.652|
> > |Meta-Llama-3-70B|0.876|0.645|0.556|0.716|0.883|0.456|0.450|
> > |Mistral-7B-v0.1|0.861|0.574|0.556|0.737|0.806|0.412|0.397|
> > |Mixtral-8x7B-v0.1|0.855|0.608|0.554|0.721|0.851|0.494|0.604|
> > |Mixtral-8x22B-v0.1|0.857|0.620|0.588|0.708|0.863|0.490|0.591|
> > ### **NQ**
> > | Rouge-L >= 0.9 | SD | SE | P(True) | Deg | NL | NE | PE |
> > |----------------|----|----|---------|-----|----|----|----|
> > |Llama-2-13b-hf|0.845|0.487|0.662|0.760|0.734|0.351|0.500|
> > |Llama-2-70b-hf|0.852|0.495|0.471|0.776|0.722|0.420|0.481|
> > |Meta-Llama-3-8B|0.812|0.431|0.420|0.702|0.818|0.410|0.595|
> > |Meta-Llama-3-70B|0.882|0.569|0.738|0.790|0.838|0.468|0.523|
> > |Mistral-7B-v0.1|0.815|0.502|0.393|0.732|0.737|0.427|0.530|
> > |Mixtral-8x7B-v0.1|0.846|0.535|0.531|0.769|0.730|0.473|0.547|
> > |Mixtral-8x22B-v0.1|0.858|0.527|0.391|0.771|0.773|0.504|0.655|

---

> ### Author Response · Authors · 2024-08-09
> **Detailed results for Rouge-L threshold of 1.0 and two ablations studies**
>
> ## Results for Rouge-L threshold of 1.0
>
> ### **CoQA**
> | Rouge-L = 1.0 | SD | SE | P(True) | Deg | NL | NE | PE |
> |----------------|----|----|---------|-----|----|----|----|
> |Llama-2-13b-hf|0.804|0.548|0.513|0.636|0.798|0.429|0.472|
> |Llama-2-70b-hf|0.807|0.546|0.464|0.615|0.799|0.421|0.461|
> |Meta-Llama-3-8B|0.767|0.495|0.528|0.730|0.685|0.420|0.515|
> |Meta-Llama-3-70B|0.801|0.563|0.644|0.617|0.788|0.387|0.467|
> |Mistral-7B-v0.1|0.815|0.561|0.557|0.651|0.784|0.413|0.460|
> |Mixtral-8x7B-v0.1|0.811|0.565|0.487|0.635|0.799|0.420|0.471|
> |Mixtral-8x22B-v0.1|0.809|0.537|0.484|0.612|0.786|0.419|0.471|
> ### **TriviaQA**
> | Rouge-L = 1.0 | SD | SE | P(True) | Deg | NL | NE | PE |
> |----------------|----|----|---------|-----|----|----|----|
> |Llama-2-13b-hf|0.893|0.546|0.551|0.756|0.914|0.465|0.509|
> |Llama-2-70b-hf|0.895|0.559|0.560|0.708|0.903|0.466|0.490|
> |Meta-Llama-3-8B|0.855|0.507|0.608|0.714|0.838|0.413|0.417|
> |Meta-Llama-3-70B|0.843|0.540|0.602|0.679|0.828|0.425|0.446|
> |Mistral-7B-v0.1|0.868|0.559|0.542|0.753|0.772|0.447|0.446|
> |Mixtral-8x7B-v0.1|0.867|0.568|0.495|0.709|0.844|0.487|0.502|
> |Mixtral-8x22B-v0.1|0.865|0.571|0.533|0.674|0.860|0.496|0.547|
> ### **SciQ**
> | Rouge-L = 1.0 | SD | SE | P(True) | Deg | NL | NE | PE |
> |----------------|----|----|---------|-----|----|----|----|
> |Llama-2-13b-hf|0.890|0.574|0.490|0.788|0.880|0.407|0.503|
> |Llama-2-70b-hf|0.888|0.676|0.577|0.732|0.861|0.453|0.526|
> |Meta-Llama-3-8B|0.881|0.602|0.544|0.734|0.881|0.464|0.652|
> |Meta-Llama-3-70B|0.876|0.645|0.556|0.716|0.883|0.456|0.450|
> |Mistral-7B-v0.1|0.861|0.574|0.556|0.737|0.806|0.412|0.397|
> |Mixtral-8x7B-v0.1|0.856|0.608|0.553|0.721|0.851|0.493|0.604|
> |Mixtral-8x22B-v0.1|0.857|0.620|0.587|0.708|0.863|0.490|0.592|
> ### **NQ**
> | Rouge-L = 1.0 | SD | SE | P(True) | Deg | NL | NE | PE |
> |----------------|----|----|---------|-----|----|----|----|
> |Llama-2-13b-hf|0.845|0.487|0.662|0.760|0.734|0.349|0.500|
> |Llama-2-70b-hf|0.853|0.494|0.469|0.776|0.721|0.418|0.480|
> |Meta-Llama-3-8B|0.813|0.429|0.418|0.703|0.819|0.409|0.595|
> |Meta-Llama-3-70B|0.883|0.569|0.738|0.791|0.838|0.467|0.523|
> |Mistral-7B-v0.1|0.815|0.502|0.392|0.732|0.737|0.427|0.530|
> |Mixtral-8x7B-v0.1|0.847|0.535|0.531|0.769|0.730|0.473|0.547|
> |Mixtral-8x22B-v0.1|0.859|0.526|0.391|0.772|0.773|0.504|0.655|
>
> ## Ablation study where the number of reference responses changes for Semantic Entropy
>
> ### **CoQA**
> | sample number | 1 | 2 | 3 | 4 | 5 | 6 | 7 | 8 | 9 | 10 |
> |---------------|---|---|---|---|---|---|---|---|---|----|
> |Llama-2-13b-hf|0.600|0.605|0.616|0.626|0.630|0.635|0.633|0.633|0.636|0.633|
> |Llama-2-70b-hf|0.605|0.624|0.625|0.629|0.628|0.625|0.623|0.622|0.623|0.621|
> |Meta-Llama-3-8B|0.571|0.579|0.585|0.598|0.601|0.600|0.606|0.609|0.608|0.599|
> |Meta-Llama-3-70B|0.604|0.597|0.599|0.604|0.607|0.608|0.609|0.609|0.608|0.608|
> |Mistral-7B-v0.1|0.572|0.596|0.612|0.617|0.624|0.626|0.626|0.628|0.630|0.627|
> |Mixtral-8x7B-v0.1|0.594|0.609|0.619|0.621|0.620|0.625|0.625|0.624|0.625|0.626|
> |Mixtral-8x22B-v0.1|0.591|0.603|0.614|0.617|0.616|0.615|0.614|0.614|0.614|0.614|
> ### **TriviaQA**
> | sample number | 1 | 2 | 3 | 4 | 5 | 6 | 7 | 8 | 9 | 10 |
> |---------------|---|---|---|---|---|---|---|---|---|----|
> |Llama-2-13b-hf|0.542|0.649|0.672|0.679|0.679|0.676|0.675|0.673|0.673|0.672|
> |Llama-2-70b-hf|0.645|0.670|0.691|0.686|0.685|0.683|0.676|0.677|0.675|0.677|
> |Meta-Llama-3-8B|0.656|0.674|0.685|0.685|0.682|0.680|0.674|0.669|0.665|0.662|
> |Meta-Llama-3-70B|0.585|0.615|0.645|0.653|0.663|0.666|0.667|0.666|0.666|0.663|
> |Mistral-7B-v0.1|0.674|0.694|0.703|0.707|0.704|0.703|0.699|0.693|0.692|0.690|
> |Mixtral-8x7B-v0.1|0.600|0.644|0.678|0.682|0.685|0.687|0.687|0.688|0.687|0.685|
> |Mixtral-8x22B-v0.1|0.615|0.655|0.677|0.688|0.692|0.693|0.692|0.691|0.688|0.686|
>
> ## Ablation study where the diversity penalty (dp) of diverse beam search changes for semantic density (using Mistral-7B)
>
> | AUROC    | dp=0.2 | dp=0.5 | dp=1.0 |
> |----------|--------|--------|--------|
> | CoQA     | 0.784  | 0.785  | 0.788  |
> | TriviaQA | 0.861  | 0.863  | 0.866  |
> | SciQ     | 0.765  | 0.766  | 0.771  |
> | NQ       | 0.678  | 0.678  | 0.680  |

---

> ### Author Response · Authors · 2024-08-14
> **Message to Reviewer koQE**
>
> Since we won't be able to post any replies after the discussion period (ending soon), please let us know if you have any further questions. Thank you!

---

> > ### Comment · Reviewer_koQE · 2024-08-14
> >
> > Thank you for providing the detailed results. As a side note, these 20 tables could have been effectively summarized in a line plot (x-axis: correctness threshold, y-axis: AUROC). Overall, I have decided to increase my score to 5.

---

> > > ### Author Response · Authors · 2024-08-14
> > > **Response to Reviewer koQE**
> > >
> > > Many thanks for your reply. We will include the line plot as you suggested in the revision.

---

> ### Comment · Reviewer_koQE · 2024-08-14
> **Final remarks**
>
> I want to emphasize once more that, while the method empirically performs very well, it lacks a solid theoretical foundation within the context of current uncertainty quantification. The prevailing theory suggests that uncertainty quantification should focus on evaluating the predictive distribution (over classes, semantic clusters, etc.), irrespective of the sampled output. To draw a parallel with the typical classification setting, one also doesn't assign different uncertainty estimates to different classes sampled from the predictive distribution – to evaluate the performance of an uncertainty estimate, one uses the argmax to determine wether the model "knows" the correct class (i.e. should be certain) or "doesn't know" the correct class (i.e. should be uncertain). This principle can be similarly applied to uncertainty estimates in NLG. It should be carefully elaborated on in the updated version of the paper.

---

> > ### Author Response · Authors · 2024-08-14
> > **Response to Reviewer koQE**
> >
> > Thanks for your final remarks. We will make sure to discuss and clarify this point in a careful way in the revised version, based on our insightful discussions. Thank you again for your time and effort!

---

### Official Review · Reviewer_kNB4 · 2024-07-13

**Soundness:** 3
**Presentation:** 3
**Contribution:** 3
**Rating:** 5
**Confidence:** 4

**Summary:**

The paper proposes a novel framework for quantifying uncertainty in LLM's responses. The authors introduce the concept of "semantic density" (SD), which measures uncertainty from a probability distribution perspective in semantic space, applicable to any pre-trained LLM without additional training. Experiments on various state-of-the-art LLMs and benchmarks demonstrate that SD outperforms existing methods in accuracy and robustness. The paper presentation is overall easy to read.

**Strengths:**

1. The paper introduces a novel method, semantic density, that quantifies uncertainty in LLM responses using semantic information.
2. The proposed framework is clear and detailed, including its theoretical foundations and implementation details.
3. The proposed method is off-the-shelf and can be applied to other scenarios.

**Weaknesses:**

1. The comparison with existing methods, although extensive, may not cover all possible alternatives. There are recent or lesser-known methods that also warrant consideration.
2. While the method shows promise for free-form question-answering tasks, its applicability and performance in other types of tasks (e.g., summarization, translation) are not explored, which limits the generalizability of the findings.

**Questions:**

1. The literature review is not comprehensive, there are lots of related works [1, 2, 3] not mentioned or adopted in the experiment.
2. The AUROC is not enough the show the strength of the proposed model. It would be more beneficial to show Area under precision-recall curve for the misclassification detection task as well.
3. The prompt construction can have more details. In Sec. 3.2, it's worth denoting the final prompt structure, since the in-context learning examples and prompt structures can make the final result different.
4. The dataset are mostly QA tasks, it seems the model can also handle other NLG tasks. Can you elaborate more on this?

[1] Ling, Chen, et al., "Uncertainty Quantification for In-Context Learning of Large Language Models." (NAACL 2024)
[2] Fadeeva, Ekaterina, et al. "LM-polygraph: Uncertainty estimation for language models." (EMNLP 2023)
[3] Duan, Jinhao, Hao Cheng, Shiqi Wang, Chenan Wang, Alex Zavalny, Renjing Xu, Bhavya Kailkhura, and Kaidi Xu. "Shifting attention to relevance: Towards the uncertainty estimation of large language models." ACL (2024).

---

> ### Author Rebuttal · Authors · 2024-08-07
>
> Comment: The comparison with existing methods, although extensive, may not cover all possible alternatives. There are recent or lesser-known methods that also warrant consideration.
>
> Response: Thanks for the suggestions. We tried to include all the mainstream uncertainty quantification methods for LLMs based on our literature review. Following your suggestion, we have done another literature search for more recent works, even including the papers that are published after the submission deadline of our current manuscript (May 22). We found another variant of semantic entropy from a recent paper published in Nature [4] (published after our submission), and we have added it into our experiments. Based on the results, the performance of the new variant of semantic entropy is similar to the original version of semantic entropy, and the proposed semantic density consistently outperforms it with different base LLMs and datasets. We will include these results in the revision.
>
> ---
>
> Comment: While the method shows promise for free-form question-answering tasks, its applicability and performance in other types of tasks (e.g., summarization, translation) are not explored, which limits the generalizability of the findings.
>
> Response: Thanks for your constructive comment. To further demonstrate the generalizability of the proposed method, we have added a summarization task with the commonly used DUC dataset. Based on the experimental results (please see detailed results in general response to all reviewers), the proposed semantic density still shows best performance compared to other counterparts, demonstrating its good generalizability. We will include these results in the revised version.
>
> ---
>
> Comment: The literature review is not comprehensive, there are lots of related works [1, 2, 3] not mentioned or adopted in the experiment.
>
> Response: Thanks for the suggestion. We have actually already included an earlier version of [1] in our literature review, we will update it into this newest version. For [2], it is a library that implements some of the existing uncertainty quantification methods for LLMs, without proposing new methods. We will include it for completeness. We will add and discuss [3] in the revised version.
>
> ---
>
> Comment: The AUROC is not enough to show the strength of the proposed model. It would be more beneficial to show Area under precision-recall curve for the misclassification detection task as well.
>
> Response: Thanks for this constructive comment. As you suggested, we have added the Area under precision-recall (AUPR) score as an additional performance metric. Please see table 1 in the attached PDF for detailed results. From the experimental results, the proposed semantic density shows consistently better performance compared to other methods, verifying its effectiveness.
>
> ---
>
> Comment: The prompt construction can have more details. In Sec. 3.2, it's worth denoting the final prompt structure, since the in-context learning examples and prompt structures can make the final result different.
>
> Response: Thanks for this constructive suggestion. To make the comparisons fair, we used the same prompt template as used in the original semantic entropy paper [5]. We will add the detailed prompt template in the revised version.
>
> ---
>
> Comment: The dataset are mostly QA tasks, it seems the model can also handle other NLG tasks. Can you elaborate more on this?
>
> Response: You are right. In principle, the proposed semantic density does not pose any restrictions on the task type, and it should work for a wide range of free-form NLG tasks. Following your previous suggestion, we have added a summarization task to demonstrate this generalizability of semantic density. We will also clarify this point in the revision.
>
> [1] Ling, Chen, et al., "Uncertainty Quantification for In-Context Learning of Large Language Models." (NAACL 2024)
> [2] Fadeeva, Ekaterina, et al. "LM-polygraph: Uncertainty estimation for language models." (EMNLP 2023)
> [3] Duan, Jinhao, Hao Cheng, Shiqi Wang, Chenan Wang, Alex Zavalny, Renjing Xu, Bhavya Kailkhura, and Kaidi Xu. "Shifting attention to relevance: Towards the uncertainty estimation of large language models." ACL (2024).
> [4] Farquhar, Sebastian and Kossen, Jannik and Kuhn, Lorenz and Gal, Yarin, “Detecting hallucinations in large language models using semantic entropy”, Nature 630, 625–630 (2024)
> [5] Lorenz Kuhn, Yarin Gal, and Sebastian Farquhar. Semantic Uncertainty: Linguistic Invariances for Uncertainty Estimation in Natural Language Generation. In The Eleventh International Conference on Learning Representations (2023)

---

> > ### Author Response · Authors · 2024-08-09
> > **Follow-up comment to Reviewer kNB4**
> >
> > As per the request by Reviewer koQE, we have now added the detailed results for all the newly added experiments, including the one suggested by your that adds a new variant of semantic entropy. Please see our follow-up comment in the general rebuttal above for detailed results. Thank you.

---

> ### Author Response · Authors · 2024-08-13
> **Message to Reviewer kNB4**
>
> Since we will not be able to post any further responses after the discussion deadline tomorrow, please feel free to let us know if you have any feedback about our rebuttal, including those newly added experiments as suggested by you. Thank you for your attention!

---

### Official Review · Reviewer_mViE · 2024-07-14

**Soundness:** 3
**Presentation:** 3
**Contribution:** 2
**Rating:** 6
**Confidence:** 3

**Summary:**

This paper addresses the problem of LLM’s lack of uncertainty metric for the response it generates. The authors propose to use semantic density to quantify such uncertainty, as it is not restricted to any specific downstream task. In particular, the approach samples reference responses, analyzes semantic relationships, and then calculates the semantic density. Experiments on four QA benchmarks of several Llama and Mistral models are conducted, including the latest Llama 3 13B and Mistral-8x22B, to demonstrate the effectiveness and the robustness of semantic density compared to prior metrics.

**Strengths:**

* The proposed metric is not restricted to any specific task but measuring the general LLM ability
* The experiments show the effectiveness of the proposed approach, outperforming the other previous metrics

**Weaknesses:**

* Only some of the llama and mistral models are tested
* The results shown in Table 1 seem to be quite close for all the models (on each task), which are not so indicative for system comparison
* No further analysis/discussions on different model sizes (7B vs. 70B), nor different architecture (MoE vs. Non-MoE)

**Questions:**

In Figure 1, different models have inconsistent performance across different tasks. Do you have any explanations?

**Limitations:**

Yes

---

> ### Author Rebuttal · Authors · 2024-08-07
>
> Comment: Only some of the llama and mistral models are tested.
>
> Response: Thanks for the comment. We want to clarify that at the time of submission (May 22), all the open-sourced LLMs released by MistralAI were included in the experiments, e.g., 'Mistral-7B', 'Mixtral-8x7B', and 'Mixtral-8x22B', and all the open-sourced Llama3 models were included in the experiments, e.g., ‘Llama-3-8B', 'Llama-3-70B’. For Llama2, there were three open-source models, i.e., 'Llama-2-7b', 'Llama-2-13b', and 'Llama-2-70b', and we have included both 'Llama-2-13b' and 'Llama-2-70b' in our experiments. As an older and less capable small-size LLM, 'Llama-2-7b’ is usually replaced by 'Mistral-7B' or ‘Llama-3-8B' in practical usages, so we did not include it in our initial experiments. We will add 'Llama-2-7b’ in the revision for completeness.
>
> ---
>
> Comment: The results shown in Table 1 seem to be quite close for all the models (on each task), which are not so indicative for system comparison
>
> Response: Thanks for the comment. Please allow us to further clarify how to interpret the results in Table 1, then we will address your concern. Each column in Table 1 represents one uncertainty quantification method, and each row shows the results for one base LLMs, on top of which we are applying those uncertainty quantification methods. In order to compare the performance among uncertainty quantification methods, we should look at the AUROC scores in each row, the higher the better. We have highlighted the best performing entry in boldface for each row. Regarding your concern here, the performance of other uncertainty quantification methods are not always consistent or close when applying to different base LLMs. The performances of the proposed semantic density (first column) are relatively more consistent across different models (though non-trivial differences can still be observed in some tasks). This actually indicates the better robustness of semantic density compared to other methods: it is able to provide consistently good performance when applying to base LLMs of varying sizes and structures. We will include this discussion in the revision.
>
> ---
>
> Comment: No further analysis/discussions on different model sizes (7B vs. 70B), nor different architecture (MoE vs. Non-MoE)
>
> Response: Thanks for the constructive comment. As suggested, we have added an empirical analysis that compares the performance gain of semantic density vs. semantic entropy on different groups of base LLMs. The results indicate that the performance gain of semantic density is not sensitive to the model size or architectures.
>
> ---
>
> Comment: In Figure 1, different models have inconsistent performance across different tasks. Do you have any explanations?
>
> Response: Thanks for the question. The reason is that these different LLMs are of different sizes and architectures, and since they are from different LLM families, their training pipelines and training data should also be different. It is therefore natural that their performances vary across different tasks, i.e., they have different capabilities, and they may be knowledgeable when answering one category of questions while ignorant of another topic. This is why we tried to include diverse LLMs: we want to evaluate the robustness of semantic density. We will add these discussions in the revision.

---

### Official Review · Reviewer_rbw2 · 2024-07-18

**Soundness:** 3
**Presentation:** 3
**Contribution:** 3
**Rating:** 7
**Confidence:** 3

**Summary:**

The paper proposes an approach for estimating uncertainty of LLM outputs. The proposed approach begins by sampling a diverse set of responses, analyses their equivalence using a NLI classification model, and then computes the semantic density using a kernel density estimate.

**Strengths:**

- The proposed approach seems sufficiently different from prior work and combines existing ideas meaningfully. The approach is convincingly validated in experiments.
- The paper is clear for the most part.
- The problem of uncertainty estimation for generation tasks is meaningful.

**Weaknesses:**

- The paper is somewhat vague in distinguishing itself from the closest prior work on Semantic Entropy - this makes its specific contribution unclear.
- Some details of the implementation of semantic density are unclear.

**Questions:**

- Sec 3.5: Please describe how the NLI model is used for computing p_c, p_n and p_e. NLI models typically accept pairs of texts, but these probabilities seem to contain three: y_*, y_i and x.
- Do I understand correctly that embedding models (discussed in Sec 3.2) are never explicitly used in Semantic Density? Instead the NLI model is relied upon for establishing equivalence between texts and for the kernel function? If yes, the current presentation comes across as a bit confusing - please consider rewriting parts of the paper to make this more explicit.
- As I understand, the proposed approach only estimates uncertainty rather than explicitly calibrating the underlying model - and yet the proposed uncertainty estimation method helps improve predictive performance in Table 1. Am I understanding correctly that the underlying models are somewhat well calibrated to begin with? To put it another way, are there generation tasks where one might expect SD to not be as reliable? For example, a summarization task. If this is a reasonable understanding, please consider discussing where SD is unlikely to work well.
- The differences highlighted against Semantic Entropy in Lines 105-107 are vague or inaccurate - SE considers responses in computing the SE(x) value so it is unclear how it is "prompt-wise", that being said, what are the advantages of obtaining uncertainty for a specific output and prompt pair? What does a "one-cut equivalence relationship" mean? Please consider highlighting differences more clearly and discussing the specific advantages/disadvantages that the differences result in.

---

> ### Author Rebuttal · Authors · 2024-08-07
>
> Response: Semantic Entropy (SE) is indeed the baseline to which we compare, aiming to overcome its  limitations using a different approach, i.e. Semantic Density (SD). Let us first clarify the difference. although the computation of SE involves analysis of multiple reference responses, only one SE score is obtained for each prompt: SE does not quantify the predictive uncertainty of each specific response. As a simple example, assume that given a prompt X, an LLM samples three different responses A, B, and C, among which A is the correct answer while B and C are incorrect. SE returns only one uncertainty score, representing the aleatoric uncertainty introduced by prompt X. In contrast,  SD returns three different uncertainty scores, one for each specific response.
>
> This difference matters because users can get different responses from the LLM even given the same prompt. In this case, the user who gets a correct response A would be given the same SE score as another user who gets an incorrect response B. Thus, users cannot differentiate the trustworthiness of different responses for the same prompt. In contrast, SD returns different semantic density scores depending on the actual answer returned. A more trustworthy response has a high  score, and thus the users can decide whether a specific response can be trusted. This fundamental difference between SE and the proposed semantic density comes from their inherently different design principles: SE is based on entropy calculation, which can only be prompt-wise, while SD is based on density estimation, which is naturally response-wise. We will add this elaborated discussion in the revision.
>
> "What does a "one-cut equivalence relationship" mean? Please consider highlighting differences more clearly and discussing the specific advantages/disadvantages that the differences result in.”
>
> As discussed in line 51-55 in the current manuscript, the "one-cut equivalence relationship" means that the original SE applies a binary measurement of the semantic relationships between two reference responses. That is, the relationship between two reference responses can only be “equivalent” or “nonequivalent”. In contrast, SD evaluates the semantic relationships among reference responses in a continuous manner: the calculation of SD estimates the degree to which the reference responses are semantically similar. Such a continuous measurement provides a more informative and accurate semantic analysis, making the resulting uncertainty score more precise and reliable. We will further clarify this point in the revision.
>
> ---
>
> Comment: Some details of the implementation of semantic density are unclear.
>
> Response: We can see you have posted specific questions following this comment, and will address them one by one below. We will add all these clarifications in the revision to make the implementation details clearer.
>
> ---
>
> Comment: Sec 3.5: Please describe how the NLI model is used for computing p_c, p_n and p_e. NLI models typically accept pairs of texts, but these probabilities seem to contain three: y_*, y_i and x.
>
> Response: Each text is a concatenation of the prompt x and response y. The input to the NLI model is one pair of such texts: the first is the concatenation of x and y_i, and the second is the concatenation of x and y_*.  This design follows from the principle that the semantic relationship between a pair of responses (y_i and y_*) needs to be evaluated in the context of the prompt (x). We will add this clarification in the revised version.
>
> ---
>
> Comment: Do I understand correctly that embedding models (discussed in Sec 3.2) are never explicitly used in Semantic Density? Instead the NLI model is relied upon for establishing equivalence between texts and for the kernel function? If yes, the current presentation comes across as a bit confusing - please consider rewriting parts of the paper to make this more explicit.
>
> Response: That is right; thanks for pointing out the need to clarify.  The SD approach aims to connect the density estimation to a semantic embedding space, which can be done either explicitly (using an embedding model) or implicitly (using a NLI model). As explained in Line 222-225, NLI models are used in the current implementation due to the lack of contextual embedding models in the existing literature.  Please note that utilization of a NLI model also assumes there exists an implicit contextual embedding space that measures the semantic relationships. We will add this clarification in Section 3 of the revised version.
>
> ---
>
> Comment: As I understand, the proposed approach only estimates uncertainty rather than explicitly calibrating the underlying model - and yet the proposed uncertainty estimation method helps improve predictive performance in Table 1. Am I understanding correctly that the underlying models are somewhat well calibrated to begin with? To put it another way, are there generation tasks where one might expect SD to not be as reliable? For example, a summarization task. If this is a reasonable understanding, please consider discussing where SD is unlikely to work well.
>
> Response: The proposed semantic density does not modify the original underlying LLM. Instead, it provides an additional uncertainty measurement for each output response. In our experiments, all the LLMs are not guaranteed to be well calibrated. Indeed, the “NL” column in Table 1 can be seen as a measurement of calibration, and it varies a lot across LLMs.  SD performs robustly across most such variance, but we have also highlighted one case (Line 277-280 in the current manuscript) where its performance (and that of other metrics as well) is negatively affected by a badly calibrated LLM. Good calibration is thus important for uncertainty estimation in general. We will clarify this limitation in the revision.
>
> On the other hand, we have added a summarization task (DUC2004) to the experiments. SD performs well in it, demonstrating generality across tasks.

---

> > ### Comment · Reviewer_rbw2 · 2024-08-11
> > **Thank you**
> >
> > Thank you for the additional results and the responses.

---

> > > ### Author Response · Authors · 2024-08-14
> > > **Reply to Reviewer rbw2**
> > >
> > > Thank you for spending time reading our rebuttal.

---

### Author Rebuttal · Authors · 2024-08-07

We want to thank all the reviewers for their valuable time and constructive comments. We have considered every comment from each reviewer, and added several experiments as suggested by reviewers. We believe the paper is in a much better form after incorporating constructive suggestions from all the reviewers. Below is a brief summary of the new experimental results we added during rebuttal. We will also use this space to display some of the detailed results:

1. We added a new summarization dataset as suggested by reviewer rbw2 and kNB4. Below shows the detailed results on Mistral-7B and Mixtral-8x7B:
| AUROC        | SD    | SE    | Deg   | NL    | NE    | PE    |
|--------------|-------|-------|-------|-------|-------|-------|
| Mistral-7B   | 0.679 | 0.537 | 0.660 | 0.673 | 0.551 | 0.549 |
| Mixtral-8x7B | 0.640 | 0.532 | 0.620 | 0.636 | 0.576 | 0.497 |

2. We added an experiment that uses area under precision-recall curve (AUPR) score as performance metric, as suggested by reviewer kNB4. Please see Table 1 in the attached PDF for detailed results.
3. We added an experiment that uses different rouge-L threshold for correctness evaluations, as suggested by reviewer koQE.
4. We added an experiment that uses different hyperparameters for the diverse beam search, as suggested by reviewer koQE.
5. We added an experiment that studies the effect of sampling numbers on semantic entropy, as suggested by reviewer koQE.
6. We added a comparison to a more recent variant of semantic entropy, as suggested by Reviewer kNB4.

---

> ### Comment · Reviewer_koQE · 2024-08-08
>
> Thank you for providing the additional results on the summarization dataset. This leads me to the following question: Your method employs an NLI model (i.e. Deberta-large-mnli) to compute the expectation of $|| v_* - v_i ||$ (Eq. 9). Given that this NLI model is trained on single-sentence pairs from the Multi-Genre Natural Language Inference corpus, how do you apply this model to multiple-sentence generations $v$?

---

> ### Author Response · Authors · 2024-08-09
> **Response to Reviewer koQE**
>
> Thanks for your comment. The motivation of most summarization benchmarks are to evaluate a model’s ability in compressing the information, so the DUC summarization task used here requires the language models to summarize a paragraph within a predefined length limit, usually no more than a single long sentence. The current NLI model we used (Deberta-large-mnli) still works reasonably in such tasks. However, we think you did bring up a good point for future work in this research direction, i.e., how should we quantify the uncertainty of a long paragraph? Recent solutions to this scenario focuses on decomposing the long paragraph into pieces of “claims” [1] or “factoid” [2], each one being at sentence level, then we can apply existing uncertainty quantification methods to each of them. This strategy makes sense since it is indeed more informative if we could quantify the uncertainty for each part when the response is a long paragraph containing multiple claims. However, it would also be interesting if we can further develop a NLI model or embedding model that can properly measure the overall semantic similarity between two long paragraphs, or we can directly ask another LLM to make this overall similarity evaluation quantitatively. This discussion may be a little out-of-scope, but we will add this discussion in the future work section. Thanks again for your inspiring comment.
>
> [1] Xin Liu, Muhammad Khalifa, and Lu Wang. “LitCab: Lightweight Language Model Calibration over Short- and Long-form Responses”. In The Twelfth International Conference on Learning Representations (ICLR), 2024
>
> [2] Farquhar, Sebastian and Kossen, Jannik and Kuhn, Lorenz and Gal, Yarin, “Detecting hallucinations in large language models using semantic entropy”, Nature 630, 625–630, 2024

---

> ### Author Response · Authors · 2024-08-09
> **Follow-up response to the general rebuttal**
>
> Due to the space limitation of initial rebuttal, we did not include all the detailed results for the newly added experiments. As per the request by Reviewer koQE, we have now added the detailed results for all the newly added experiments. The detailed results for experiment 3, 4 and 5 are added in the comments replying to Reviewer koQE. The detailed results for experiment 6 are directly included below:
>
> ## Results for the modified version of semantic entropy [1]:
>
> | AUROC         | CoQA  | TriviaQA | SciQ  | NQ    |
> |---------------|-------|----------|-------|-------|
> | Llama-2-13B   | 0.587 | 0.726    | 0.576 | 0.576 |
> | Llama-2-70B   | 0.570 | 0.696    | 0.596 | 0.555 |
> | Llama-3-8B    | 0.606 | 0.736    | 0.587 | 0.582 |
> | Llama-3-70B   | 0.571 | 0.694    | 0.582 | 0.563 |
> | Mistral-7B    | 0.572 | 0.734    | 0.613 | 0.568 |
> | Mixtral-8x7B  | 0.567 | 0.699    | 0.576 | 0.612 |
> | Mixtral-8x22B | 0.562 | 0.690    | 0.592 | 0.565 |
>
> Its performance is comparative to the original semantic entropy [2], and the proposed semantic density still outperforms it consistently across all the base LLMs and tasks.
>
> [1] Farquhar, Sebastian and Kossen, Jannik and Kuhn, Lorenz and Gal, Yarin, “Detecting hallucinations in large language models using semantic entropy”, Nature 630, 625–630, 2024
>
> [2] Lorenz Kuhn and Yarin Gal and Sebastian Farquhar, “Semantic Uncertainty: Linguistic Invariances for Uncertainty Estimation in Natural Language Generation”, The Eleventh International Conference on Learning Representations (ICLR), 2023

---

### Decision · Program_Chairs · 2024-09-25

**Decision:**

Accept (poster)

**Comment:**

The paper proposes an approach based on kernel density estimation techniques in the semantic space for uncertainty quantification in text generation for large language models. The paper evaluated on question answering data sets, and during the rebuttal process, added results on summarization data sets as well. The reviewers agreed that the approach is principled and the emprical results are strong across benchmarks.

One concern from reviewer koQE is that semantic density lacks a solid theoretical foundation within the context of current uncertainty quantification. However, I think in practical terms, calibration to quantify the confidence of a sampled answer (e.g., at temperature 0) is useful, and the evaluation metric of AUROC also reflects this goal.